# Cells must express components of the planar cell polarity system and extracellular matrix to support cytonemes

**Hai Huang, Thomas B Kornberg\***

Cardiovascular Research Institute, University of California, San Francisco, San Francisco, United States

**Abstract** Drosophila dorsal air sac development depends on Decapentaplegic (Dpp) and Fibroblast growth factor (FGF) proteins produced by the wing imaginal disc and transported by cytonemes to the air sac primordium (ASP). Dpp and FGF signaling in the ASP was dependent on components of the planar cell polarity (PCP) system in the disc, and neither Dpp- nor FGF-receiving cytonemes extended over mutant disc cells that lacked them. ASP cytonemes normally navigate through extracellular matrix (ECM) composed of collagen, laminin, Dally and Dally-like (Dlp) proteins that are stratified in layers over the disc cells. However, ECM over PCP mutant cells had reduced levels of laminin, Dally and Dlp, and whereas Dpp-receiving ASP cytonemes navigated in the Dally layer and required Dally (but not Dlp), FGF-receiving ASP cytonemes navigated in the Dlp layer, requiring Dlp (but not Dally). These findings suggest that cytonemes interact directly and specifically with proteins in the stratified ECM.

**\*For correspondence:**
tkornberg@ucsf.edu

**Competing interests:** The authors declare that no competing interests exist.

## Introduction

The language of **development** has a small vocabulary of signaling proteins that consists in part of Fibroblast growth factor (FGF) and Bone morphogenic proteins such as Drosophila Decapentaplegic (Dpp). This language may be used in most or all metazoan organs. Studies of Drosophila, chick, zebrafish, and cultured human cells show that the signaling proteins that regulate development are transported along actin-based filopodia (cytonemes) and exchange at synapses where the cells that produce them contact the cells that receive and respond to them (*Roy et al., 2014* and reviewed in *Kornberg and Roy, 2014*; *Pröls et al., 2016*). The large distances between the source and receiving cells in some of these contexts (as much as 100 μm in the wing disc and 150 μm in the chick limb bud) highlights the question that this work investigates - how cytonemes extend to reach their targets.

The cytonemes we characterized were made by the ASP, a tracheal tube that develops in the third instar larva under the influence of Dpp and FGF that are produced in the wing disc (*Roy et al., 2011*; *Sato and Kornberg, 2002*). The cytonemes that mediate the exchange of these proteins contain the Dpp receptor Thickveins (Tkv) or the FGF receptor Breathless (Btl), extending from the basal surface of the ASP cells and synapsing with Dpp- or FGF-producing disc cells, respectively (*Roy et al., 2014*). The ASP lies underneath the basement membrane that envelops the wing disc (*Guha et al., 2009*), and although the space they traverse has not been analyzed, it presumably has characteristics of prototypical extracellular matrix (ECM, reviewed in *Broadie et al., 2011*). The number and distribution of ASP cytonemes depend on the production of Dpp and FGF in the disc and on their respective receptors in the ASP, but it is not known whether the cells between the producing and receiving cells (henceforth called 'intermediate cells') also contribute to cytoneme-mediated signaling. Possible candidates that might have roles in these intermediate cells that we tested

**eLife digest** The embryos of animals develop in a controlled manner that ensures that their tissues and organs form properly and at the right time. These processes depend on molecules called morphogens that are distributed throughout the embryo in specific ways and that are dispersed via extensions that protrude from the surfaces of cells. These extensions, called cytonemes, transport the morphogens across the distances that separate cells and transfer these molecules to target cells via direct contact. However, it was not known how cytonemes navigate to their targets.

The fruit fly *Drosophila* is commonly used to investigate how animals develop organs and tissues. Previous studies have shown that the development of one of the fly's organs – the air sac primordium –relies on morphogens transported by cytonemes.Now, Huang and Kornberg reveal that these cytonemes navigate to their targets by using the composition of the mesh-like framework – referred to as the extracellular matrix – that surrounds animal tissues as a guide. Further experiments showed that the extracellular matrix between the cells that produce the morphogens and the cells of the air sac primordium is roughly arranged into layers. These layers contain different molecules and the cytonemes navigate within specific layers.

These findings reinforce the idea that the extracellular space is organized and regulated, and show that the extracellular matrix is essential for developmental signaling. Future challenges include understanding how the layers of the extracellular matrix form and how information is encoded in these layers for the cytonemes to decipher as they navigate to their targets.

include components of the planar cell polarity (PCP) system, heparan sulfate proteoglycans (HSPGs) and integrins.

PCP is an aspect of cell polarity that establishes a singular, shared bipolar orientation across an epithelial sheet (reviewed in *Goodrich and Strutt, 2011*). In the insect cuticle, it is responsible for the coordinated and consistent orientation of hairs and bristles. PCP is also manifested in the asymmetric subcellular localization of proteins such as Frizzled (Fz, a seven-pass transmembrane protein), Dishevelled (Dsh) and Diego (Dgo, cytosolic proteins) to one side and Van Gogh (Vang, a four-pass transmembrane protein) and Prickle (Pk, a cytosolic protein) to the other. Fz, Dsh, Dgo, Vang and Pk are constituents of the core PCP pathway in Drosophila, and all are required for planar orientation and polarization. Absent any one and the hairs and bristles lack normal polarity. In Drosophila genetic mosaics, cells to one side of PCP mutant cells also have abnormal planar polarity (*Casal et al., 2002*; *Taylor et al., 1998*; *Vinson and Adler, 1987*), leading to the idea that PCP involves a system of cell-cell interactions that coordinate the polarity of neighboring cells and propagate orientation long-range.

PCP components also have other roles. Studies of cells deficient for components of the PCP system in Drosophila (*Djiane et al., 2005*; *Harumoto et al., 2010*) and mouse (*Tao et al., 2009*; *Vandenberg and Sassoon, 2009*) report loss of several features of normal apical-basal polarity. PCP components have been implicated in the deposition and remodeling of the ECM (*Dohn et al., 2013*; *Goto et al., 2005*; *Williams et al., 2012a, 2012b*), axon guidance (*Fenstermaker et al., 2010*; *Mrkusich et al., 2011*; *Shafer et al., 2011*; *Gombos et al., 2015* and reviewed in *Yam and Charron, 2013*) and cell migration (*Carreira-Barbosa et al., 2003*; *Dohn et al., 2013*; *Roszko et al., 2015*; *Tatin et al., 2013*).

The glypicans Dally and Dally-like (Dlp) are components of the ECM that contribute essential functions to signaling (*Han et al., 2004*; *Fujise et al., 2003*; *Han et al., 2005*; *Lin and Perrimon, 1999*; *Yan and Lin, 2007*; *Yan et al., 2010*). They are glycophosphatidylinositol (GPI)-anchored and are modified with heparan sulfate glycosaminoglycan (GAG) chains. Although an intracellular function has been reported for Dlp in Hh-producing cells (*Callejo et al., 2011*), it is assumed that both Dally and Dlp function externally. Models proposed for the roles of Dally and Dlp include binding as co-receptors of signaling proteins (*Fujise et al., 2003*; *Kim et al., 2011*; *Lin and Perrimon, 1999*) and participating in either a process of surface diffusion that involves repeated cycles of signaling protein-HSPG binding, dissociation, sliding and re-association with adjacent HSPG binding sites (*Han et al., 2004*; *Schlessinger et al., 1995*) or repeated cycles of ligand-HSPG binding, endocytic

uptake, and transcytosis (*Yan and Lin, 2009*). Alternatively, the recent report that cytonemes do not pass over patches of cells that are deficient for GAG-modified HSPGs suggests that impairment of cytoneme-mediated transport may account for the mutant phenotypes (*Bischoff et al., 2013*).

Integrin function is required for axonal pathfinding, TGF-β signaling and for interactions of migrating cells and cell protrusions with the ECM (*Dominguez-Giménez et al., 2007*; *Han et al., 2012*; *Myers et al., 2011*; *Robles and Gomez, 2006*; *Vuoriluoto et al., 2008* and reviewed in *Munger and Sheppard, 2011*). Although the basis for these roles has not been defined, we investigated whether integrins have an essential functional role in cytoneme-mediated signaling in the ASP system.

## Results

### Development of the air sac primordium requires components of the PCP system

Clonal analysis of mutant wing disc cells defective for the PCP system revealed both cell autonomous and cell non-cell-autonomous effects (*Adler et al., 2000*; *Casal et al., 2002*; *Taylor et al., 1998*; *Tree et al., 2002*). To investigate whether PCP mutants also affect the cytonemes that interact with the wing disc, we examined ASP morphology in PCP mutant third instar larvae. The ASP grows laterally from the transverse connective (TC) across the disc toward the cells that express FGF, and at the wandering stage (late L3), the ASP extends across the band of Dpp-expressing cells (*Figure 1A–C*). The late L3 ASP has a characteristic narrow proximal stalk, bulbous medial region and rounded distal tip. Many cytonemes emanate from its periphery, with long lateral ones that extend dorsally to the Dpp-expressing disc cells, and the longer ones at the tip that reach postero-laterally to the FGF-expressing disc cells. In *pk* and *Vang* (*Figure 1D–F*), *dachsous (ds)* and *fat (ft)* mutants (*Figure 1— figure supplement 1A–C*), and in larvae that express *misshapen, dRhoA, drok, dRac1, multiple wing hair,* or *Leukocyte-antigen-related-like* RNAi (*Table 1*), ASP morphology was abnormal. Although the abnormalities of the *pk* and *Vang* mutant ASPs shown in *Figure 1E and F* are among the most extreme that were observed, all mutant ASPs with these genotypes were abnormal. We do not make inferences about the roles of the genes for which mutant phenotypes were not observed on the screen. We investigated the phenotypes of the *pk* and *Vang* mutants further.

In *pk* and *Vang* mutants, growth was stunted, the morphology of every ASP was abnormal (approximately 10% were duplicated), and the ASP cytonemes were less abundant and shorter than normal (*Figure 1E,F* and *Table 2*). These phenotypes implicate Pk and Vang in ASP development but do not distinguish between requirements in ASP cells, in signal-producing disc cells or in the intermediate disc cells over which the cytonemes extend. In order to discriminate between these alternatives, we reduced pk and Vang function by expressing *pkRNAi* and *VangRNAi* constructs specifically in tracheal or disc cells. Expression of the RNAi constructs in tracheal cells did not affect either ASP morphology or ASP cytonemes (*Figure 2A,B*). In contrast, ASP morphology was abnormal and cytonemes were reduced in number and length when *pkRNAi* or *VangRNAi* were expressed in the dorsal region of the wing disc where the ASP is located (*Figure 2F,G*). Similar results were obtained by ectopic over-expression of *pk* and *Vang* in the dorsal wing disc (*Figure 2—figure supplement 1*), consistent with previous studies showing that loss- and gain-of-function conditions for *pk* and other PCP genes have similar effects on planar polarity (*Adler et al., 2000*; *Casal et al., 2002*; *Taylor et al., 1998*; *Tree et al., 2002*; *Vinson and Adler, 1987*). Over expression of either *fz* or *fmi* had no apparent effect on ASP morphology or development (*Figure 2—figure supplement 1*). Expression of RNAi constructs directed against other genes of the PCP system (*Table 1*) identified *Grunge (Gug)* to be another candidate function that is required specifically in disc cells (*Figure 2C,H*). *Gug* encodes the Drosophila homolog of the Atrophin co-repressor (*Erkner et al., 2002*; *Zhang et al., 2002*) and has been proposed to function in the Fat/Dachsous arm of the PCP system (*Fanto et al., 2003*). RNAi directed against several other genes that are required for PCP function (e.g., *dRhoA, mwh, Lar*) resulted in abnormal ASP phenotypes after expression in either the ASP or disc (*Table 1*).

To examine the genetic requirements at cellular resolution, and specifically to determine whether cytonemes are influenced by the cells they encounter as they navigate to the cells they target, we generated mitotic recombination clones of *pk* and *Vang* mutant cells in wing discs. Ten *pk* and nine

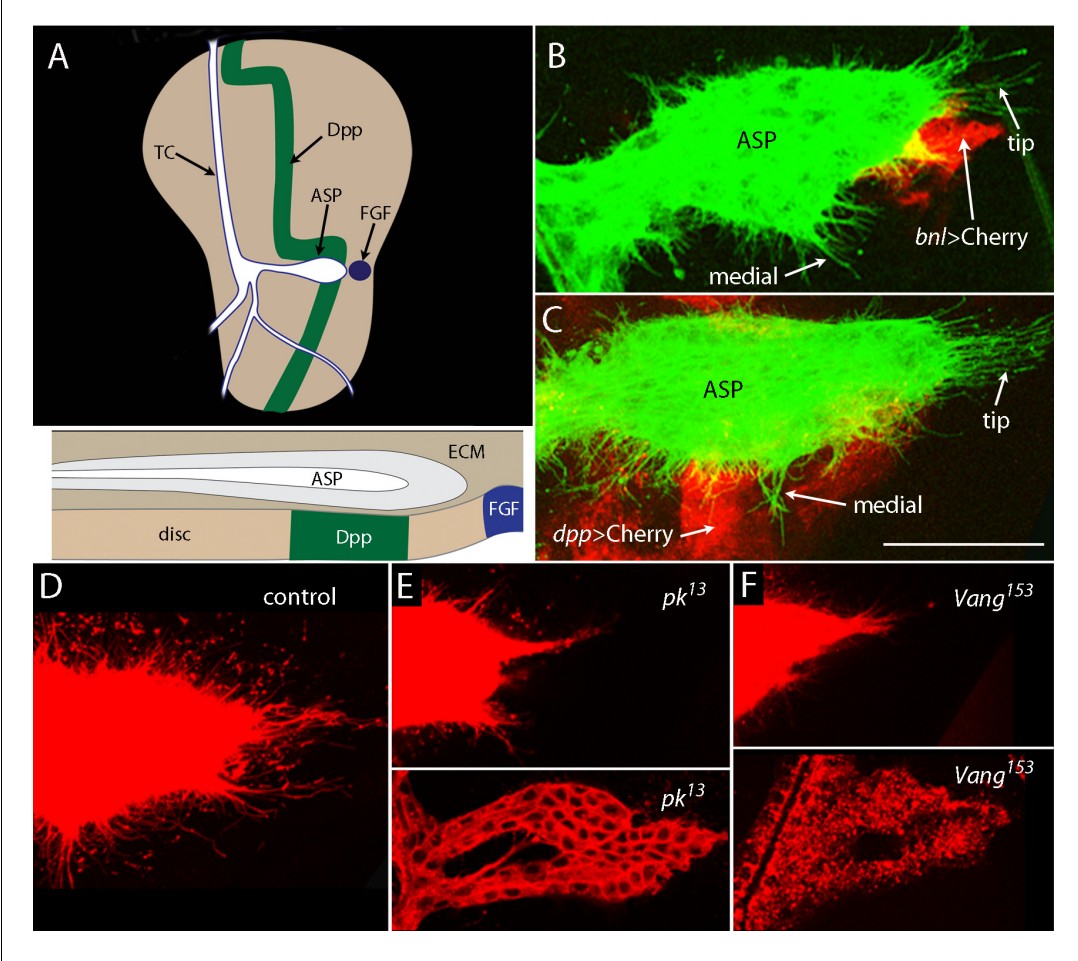

**Figure 1.** ASP cytonemes depend on *Prickle* and *Van Gogh*. (A) The drawing of a wing disc of a wandering stage third instar larva showing branches of disc-associated trachea (white, outlined in blue) with transverse connective (TC) and air sac primordium (ASP) indicated, and with the Dpp- (green) and FGF-expressing cells (blue) indicated. Sagittal section with ECM also indicated, below. (B,C) Unfixed preparations of wing disc expressing CD2:GFP driven by *btl-LHG*, and mCherry driven by *bnl-Gal4* (B) and mCherry driven by *dpp-Gal4* (C). (D–F) ASPs marked by Cherry-CAAX (driven by *btl-LHG*) in control (D) and *pk* (E) and *Vang* (F) mutants typify normal ASPs with many cytonemes extending from the entire periphery (D) and morphologically abnormal ASPs with few cytonemes in the mutants. Bottom panels show extreme examples of duplicated ASPs in the mutants. Scale bar: 30 µm.

The following source data and figure supplements are available for figure 1:

**Source data 1.** Numbers of ASP cytonemes in control, *pk* mutants and *Vang* mutants.

**Figure supplement 1.** Components of planar cell polarity modulate the activity of cytoneme-mediated signaling.

**Figure supplement 1—source data 1.** Numbers of ASP cytonemes in *ds* and *ft* mutants.

*Vang* clones were identified near the lateral side of the ASP or near the ASP tip; none of the clonal areas were traversed by cytonemes. *Figure 3* shows eight examples. Cytonemes were present at the periphery of mutant clones, but no cytonemes were found inside mutant territory. These results suggest that Pk and Vang are essential in 'intermediate' disc cells in order for cytonemes to extend over their surface.

## Dpp and FGF signaling requires Prickle and Van Gogh

Our previous studies showed that cytonemes that extend from the ASP to the wing disc are essential for Dpp and FGF signaling in the ASP (*Roy et al., 2014*). We therefore sought to determine if

**Table 1.** Expression of RNAi directed against PCP component genes.

| PCP genes | btl-Gal4 | ap-Gal4 |
|---|---|---|
| *frizzled* | – | – |
| *disheveled* | – | – |
| *Van Gogh* | – | reduced |
| *prickle* | – | small |
| *flamingo* | – | – |
| *diego* | – | – |
| *fat* | – | – |
| *dachsous* | – | – |
| *four-jointed* | – | – |
| *Grunge* | – | small |
| *Casein Kinase 1ε (discs overgrown)* | – | – |
| *G protein o α47A (brokenheart)* | – | – |
| *Dishevelled Associated Activator of Morphogenesis* | – * | – |
| *dRac1* | small | – |
| *dRhoA* | small | small |
| *misshapen* | abnormal | – |
| *Rho Kinase (drok)* | small | – |
| *multiple wing hair (mwh)* | 2 ASP | 2 ASP |
| *nemo* | – | – |
| *mushroom body defect* | – | – |
| *kugelei* | – | – |
| *Leukocyte-antigen-related-like (Lar)* | no ASP | abnormal |

"–" normal ASP.

* lethal; repression by Gal80[ts] relieved for 18 hr at third instar.

signaling in the ASP was also affected by conditions that reduced expression of *pk* and *Vang*. To monitor Dpp and FGF signaling in the ASP, we used a sensor that generates green fluorescence in cells that actively transduce Dpp (Dad-GFP; *Ninov et al., 2010*), and an antibody that detects dpERK, which accumulates in cells that actively transduce FGF (*Reich et al., 1999*). Under normal conditions, cells in the medial region of the ASP activate Dpp signal transduction (*Figure 4A* and *Roy et al., 2014*) and cells at the tip activate FGF signal transduction (*Figure 4D* and *Roy et al., 2014*; *Sato and Kornberg, 2002*). In *pk* and *Vang* mutants, Dpp and FGF signal transduction in the ASP were reduced (*Figure 4B,C,E,F*), confirming the importance of these functions for signaling in the ASP. Cells in mutant ASPs appeared to be larger and reduced in number compared to controls, and their morphology appeared to be abnormal.

## The constitution of the ECM is dependent on *prickle* and *Van Gogh*

The wing disc is enveloped by a collagen-containing ECM, and because the ASP is situated between the basal surface of the disc and ECM (*Guha et al., 2009*), the ASP cytonemes do not normally penetrate from outside the ECM to reach the disc. Rather, they navigate along, or at some distance from the basal surface of the disc cells, and they presumably encounter the ECM in a manner that has not been defined. We previously observed that the structure of the ECM is sensitive to both increased and reduced Mmp2 expression, and that dis-regulation of Mmp2 reduces ASP growth and signaling and perturbs morphogenesis (*Guha et al., 2009*). Expression of *dallyRNAi* or *dlpRNAi* in the disc perturbed ASP morphology and reduced the number of ASP cytonemes (*Figure 2I,J*), an indication that the Dally and Dlp HSPGs are essential ECM components for ASP morphogenesis and

**Table 2.** Numbers of ASP cytonemes in PCP system, HSPG and integrin pathway, mutants.

**ASP cytonemes in *prickle* and *Van Gogh* mutants**

| Genotype | # cytonemes per μm* | t-test[†] |
|---|---|---|
| btl-LHG,lexO-Cherry:CAAX/+ | 0.66 ± 0.07 | p value |
| pk^pk-sple-13/pk^pk-sple-13; btl-LHG,lexO-Cherry:CAAX/+ | 0.28 ± 0.07 | 2.09E-05 |
| Vang^153/Vang^153; btl-LHG,lexO-Cherry:CAAX/+ | 0.35 ± 0.05 | 3.65E-05 |
| btl-Gal4,UAS-CD8:mCherry/+;dally^80/dally^80 | 0.28 ± 0.05 | 9.24E-06 |
| btl-Gal4,UAS-CD8:mCherry/+;dlp^A187/dlp^A187 | 0.46 ± 0.06 | 9.85E-04 |

**ASP cytonemes in *wingblister* and *multiple edematous wings* mutants**

| Genotype | # cytonemes per μm* | t-test[†] |
|---|---|---|
| btl-LHG,lexO-Cherry:CAAX/+ | 0.68 ± 0.14 | P value |
| mew^M6/+;; btlLHG,lexO-Cherry-CAAX/+ | 0.57 ± 0.09 | 0.19[‡] |
| wb^3/+; btlLHG,lexO-Cherry-CAAX/+ | 0.58+0.08 | 0.21[‡] |
| mew^M6/+; wb^3/+; btlLHG,lexO-Cherry-CAAX/+ | 0.30 ± 0.09 | 1.02E-03 |

\* cytonemes were counted around approximately one-half the perimeter of the ASP in images generated as projection stacks from approximately 20–25 optical sections.

[†] significance for each genotype was calculated against the *btl>Cherry:CAAX* controls.

[‡] not significant.

signaling. Expression of these RNAi constructs had no effect when expressed in the ASP (*Figure 2D, E*).

To investigate if there is a link between Pk and Vang and the ECM, we characterized the Dlp, Dally and laminin in *pk* and *Vang* mutants. Using antibody directed against Dlp, we found that the level of Dlp was reduced approximately 60% in the areas over *pk* and *Vang* mutant clones (*Figure 5A–D*). The correspondence between the clone borders and areas of reduced Dlp appeared to be less than a cell diameter. To monitor Dally, we analyzed a Dally:YFP protein trap allele (because antibody against Dally was not available). As indicated by YFP fluorescence, Dally was present at similar levels over the wing disc, but fluorescence was reduced approximately 45% in the *engrailed* expression domain in discs in which either *pkRNAi* or *VangRNAi* was expressed in the posterior compartment under the control of *engrailed*-Gal4 (*Figure 5E–G*). The reduction in Dally:YFP fluorescence appeared to coincide with the anterior-posterior compartment border.

*pk* loss-of-function clones also had reduced levels of laminin (*Figure 5H,I*). The correspondence between the clone borders and areas of reduced laminin was not as tight as was noted for Dlp and Dally. Together, these observations indicate that pk and Vang are necessary to maintain normal levels of Dally, Dlp and laminin in the ECM.

To investigate the link between the PCP components and levels of ECM proteins, we monitored Dally and Dlp transcripts by Q-PCR. We did not detect changes in amounts of either *Dally* or *Dlp* RNA when normal discs were compared to *pk* or *Vang* mutant discs (*Figure 5—figure supplement 1*). We also monitored cell components that localize to either the apical (atypical PKC; aPKC) or basolateral (Discs large; Dlg) compartments (*Figure 5—figure supplement 2*). Whereas cells in mutant *Vang* clones appeared to retain normal distributions of Dlg, which is consistent with the normal morphological appearance of the columnar disc cells, levels of aPKC were elevated and extended more basally than normal. The apparent downregulation of aPKC by the PCP system, which has been reported previously in studies of the Drosophila eye (*Djiane et al., 2005*) and wing (*Harumoto et al., 2010*), indicates that some aspects of the apical/basal polarity are PCP-dependent. We do not know how the amounts of Dally and Dlp protein in the ECM are controlled or how the PCP components Pk and Vang, which concentrate apically, might influence the composition of the basal ECM. Possibly, mutant cells fail to export the HSPGs, fail to retain HSPGs that they or their neighbors make, or do not effectively sort or localize the HSPGs between their apical and basal compartments.

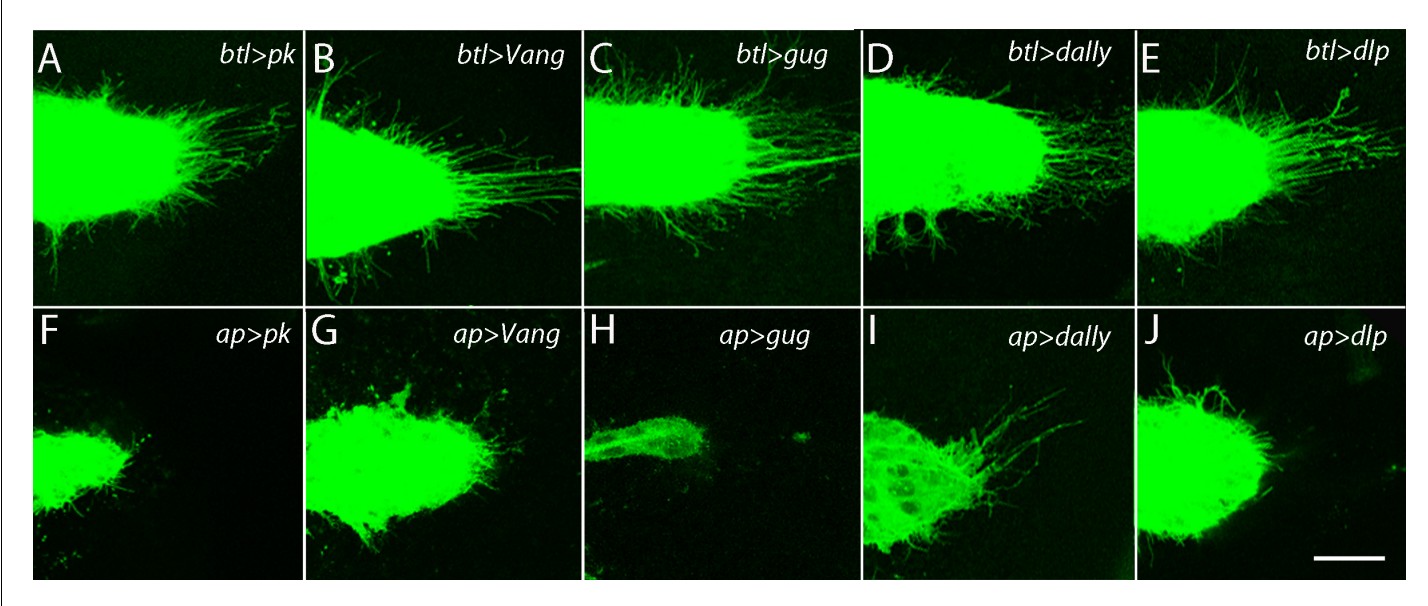

**Figure 2.** The ASP depends on the components of the PCP system in the wing disc and not in the ASP. ASPs marked by membrane-tethered GFP and expressing RNAi constructs in either the ASP (driven by *btl-Gal4*) or the dorsal compartment of the wing disc (driven by *ap-Gal4*) and directed against the indicated genes. Genotypes: (A) *btl-Gal4 UAS-CD8:GFP/+; UAS-pkRNAi/+;* (B) *btl-Gal4 UAS-CD8:GFP/UAS-VangRNAi;* (C) *btl-Gal4 UAS-CD8:GFP/+; UAS-gugRNAi/+;* (D) *btl-Gal4 UAS-CD8:GFP/UAS-dallyRNAi; UAS-dallyRNAi/+;* (E) *btl-Gal4 UAS-CD8:GFP/+; UAS-dlpRNAi/+;* (F) *ap-Gal4/+; btl-LHG lexO-CD2:GFP/UAS-pkRNAi;* (G) *ap-Gal4/UAS-VangRNAi; btl-LHG lexO-CD2:GFP/+;* (H) *ap-Gal4/+; UAS-gugRNAi/btl-LHG lexO-CD2:GFP;* (I) *ap-Gal4/UAS-dallyRNAi; btl-LHG lexO-CD2:GFP/UAS-dallyRNAi;* (J) *ap-Gal4/+; btl-LHG lexO-CD2:GFP/UAS-dlpRNAi.* Scale bar: 25 μm.

The following source data and figure supplements are available for figure 2:

**Figure supplement 1.** Dependence of ASP cytonemes on *Prickle* and *Van Gogh* expression.

**Figure supplement 1—source data 1.** Numbers of ASP cytonemes in flies over-expressing the PCP components.

## Dpp signaling depends on *dally* but FGF signaling depends on *dlp*

To investigate whether Dally and Dlp are required for cytoneme-mediated Dpp and FGF transport and for Dpp and FGF signal transduction in the ASP, we characterized the morphology of the ASP in

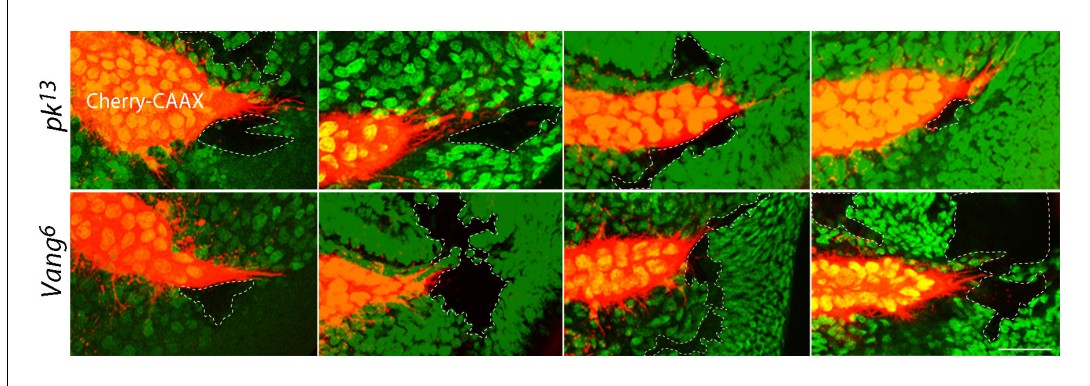

**Figure 3.** Extension of ASP cytonemes depends on *Prickle* and *Van Gogh* in underlying wing disc cells. *pk13* (upper panels) and *Vang6* (lower panels) mutant clones in wing discs with ASPs marked with Cherry-CAAX. The mutant clones are outlined with dotted lines and do not express GFP. Scale bar: 25 μm.

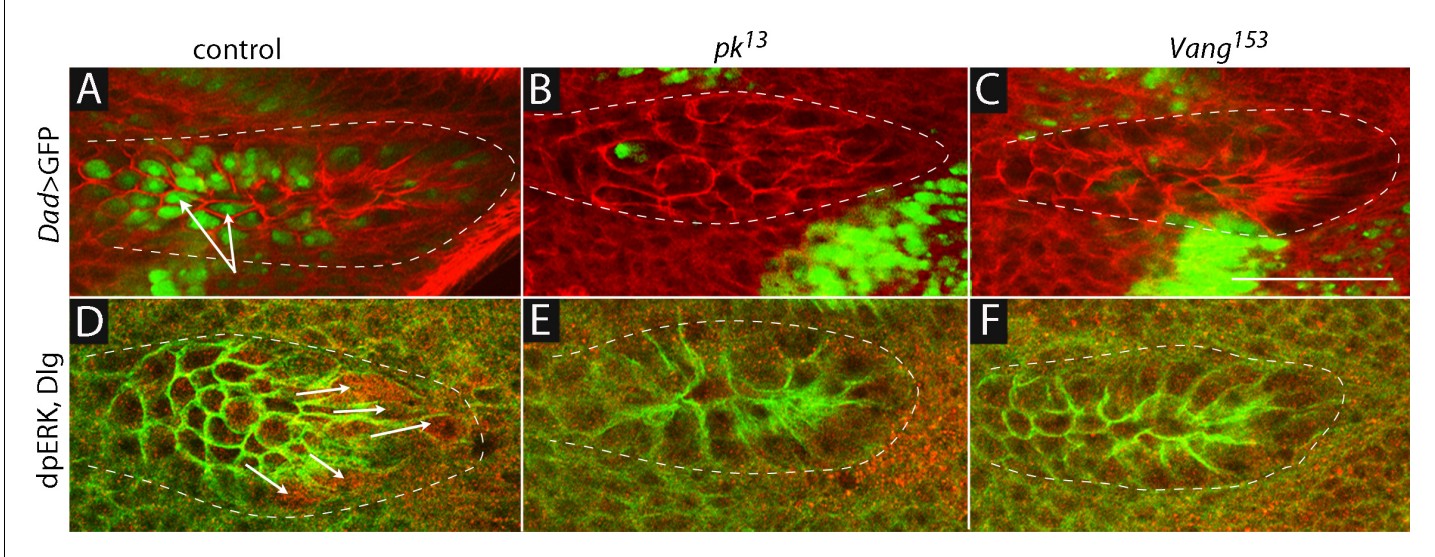

**Figure 4.** Dpp and FGF signaling in the ASP depend on *Prickle* and *Van Gogh*. (**A–C**) *Dad*-GFP expression in control (*+/+; Dad-GFP/+*), *pk* mutant (*pk¹³/pk¹³; Dad-GFP/+*) and *Vang* mutant (*Vang¹⁵³/Vang¹⁵³; Dad-GFP/+*) ASPs marked with α-Dlg. (**D–F**) dpERK staining in control, *pk¹³* mutant and *Vang¹⁵³* mutant flies. The mutant ASPs (**B,C,E,F**) were selected because their morphology most approximated normal. Arrows point to the ASP cells expressing Dad-GFP (**A**) and dpERK (**D**). Scale bar: 50 μm.

*dally* and *dlp* mutants. Whereas expression of membrane-tethered mCherry (*btl>CD8:Cherry*) in the ASP of controls revealed a large bulbous organ with cytonemes extending from both the tip (arrow) and lateral (arrowhead) regions (*Figure 6A*), expression of mCherry in *dally⁸⁰* and *dlp^A187* mutants revealed ASPs that were reduced in size and abnormally shaped (*Figure 6B,C*). In the *dally⁸⁰* ASP, cytonemes extended primarily postero-laterally from the tip; in contrast, long cytonemes extended only along the dorsal/ventral axis from the lateral surface in the *dlp^A187* ASP.

Signaling in the mutant ASPs was also affected, but the effects were specific to either the Dpp or FGF signaling pathway. Loss of Dally reduced Dpp signaling, but it did not alter FGF signaling (*Figure 6D,E,G,H*). In contrast, loss of Dlp reduced FGF signaling but not Dpp signaling (*Figure 6F, I*). *Figure 6J* presents Rose plots that show the number and orientation of ASP cytonemes in control, *dally⁸⁰* and *dlp^A187* genotypes. This analysis confirms that whereas cytonemes extended postero-laterally from the ASP tip and along the dorsal/ventral axis from the medial ASP in approximately equal numbers under normal conditions, the dorsal-oriented cytonemes that mediate Dpp signaling were primarily reduced in *dally⁸⁰* mutants and the postero-laterally oriented cytonemes were primarily reduced in *dlp^A187* mutants. Thus, the absence of lateral cytonemes in the *dally* loss-of-function condition and the absence of tip cytonemes in the *dlp* loss-of-function condition correlate with the reduction of Dpp and FGF signaling, respectively. The presence of tip cytonemes in the *dally* loss-of-function condition correlates with FGF signaling in the ASP, and the presence of lateral cytonemes in *dlp* loss-of-function conditions correlates with Dpp signaling.

To examine these differential effects of the *dally* and *dlp* mutants in finer detail, we generated *dally* and *dlp* mutant clones in the wing disc and monitored ASP cytonemes that were marked with either fluorescent FGF receptor (*btl>Btl:Cherry*) or fluorescent Dpp receptor (*btl>Tkv:Cherry*). Whereas cytonemes marked with Btl:Cherry extended over *dally* mutant clones and were apparently unchanged relative to controls, they did not extend over *dlp* mutant cells (*Figure 6K,L*). In contrast, cytonemes marked with Tkv:Cherry extended over *dlp* mutant cells without apparent perturbation, but they did not extend over *dally* mutant cells (*Figure 6M,N*). These results confirm the distinct requirements that Dpp signaling and Tkv-containing cytonemes have for *dally*, and that FGF signaling and Btl-containing cytonemes have for *dlp*.

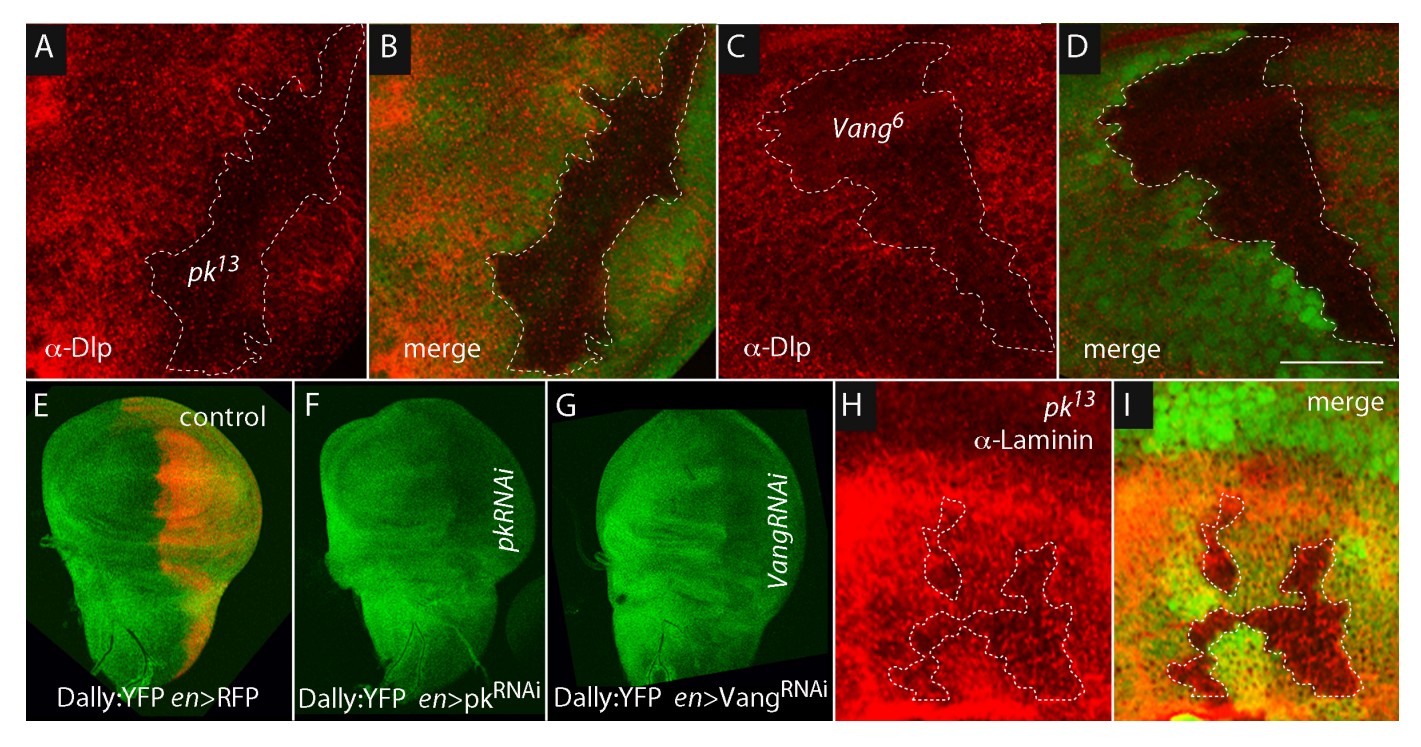

**Figure 5.** Dally and Dlp depend on *Prickle* and *Van Gogh*. Levels of Dlp were reduced in *pk13* (A,B) and *Vang6* (C,D) mutant clones (outlined with dashed white lines and lacking green fluorescence). (E–G) Levels of fluorescence in the wing discs of the protein trap Dally:YFP line were relatively uniform in controls (*en-Gal4 UAS-RFP/+; dally:YFP/+*), but were reduced specifically in the posterior compartment in the presence of *pkRNAi* (*en-Gal4/+; dally:YFP/UAS-pkRNAi*) and *VangRNAi* (*en-Gal4/UAS-VangRNAi; dally:YFP/+*). (H,I) Levels of Laminin detected by α-Laminin antibody staining (red) were reduced in *pk13* mutant clones (outlined with dashed white line and lacking green fluorescence). Scale bar: 50 μm.

The following source data and figure supplements are available for figure 5:

**Source data 1.** Quantification of fluorescence intensity of α-Dlp staining, Dally:YFP and α-Laminin staining.

**Figure supplement 1.** Quantification of *Dally* and *Dlp* transcripts in *Prickle* and *Van Gogh* mutant wing discs.

**Figure supplement 1—source data 1.** Threshold cycles for *dally, dlp* and *actin* transcripts in *wild-type, pk* mutants and *Vang* mutants.

**Figure supplement 2.** Abnormal apical/basal polarity in *Van Gogh* mutant cells.

## Cytonemes navigate in a stratified ECM

To investigate how the Tkv- and Btl-containing cytonemes might engage the Dally and Dlp HSPGs, respectively, we examined the relative locations of these ECM proteins in wing disc preparations. Simultaneous expression of Tkv:GFP and Btl:Cherry in the ASP marked the Tkv- and Btl-containing cytonemes, and sagittal projections of serial confocal sections (*Figure 7A*) and frontal sections (*Figure 7B,C*) of a specimen isolated from an early to mid L3 stage larva showed that these two types of cytonemes are separated in the space over the wing disc: the Tkv-containing cytonemes traverse a region further from the basal surface of the disc than the Btl-containing ones. Both the Tkv-containing and Btl-containing cytonemes extend from the tip at this stage (*Roy et al., 2014*). Similar preparations imaged to identify Dally and Dlp (*Figure 7D*), and laminin, collagen and Dlp (*Figure 7E*), showed that these proteins were stratified above the basal surface of the disc epithelium. Collagen was present in a discrete, distal-most layer. Laminin was also most prominent distally but it did not extend as far from the disc surface as collagen. Dally was also prominent distally, but Dally did not extend as far from the disc surface as collagen or laminin. Dlp was most prominent in

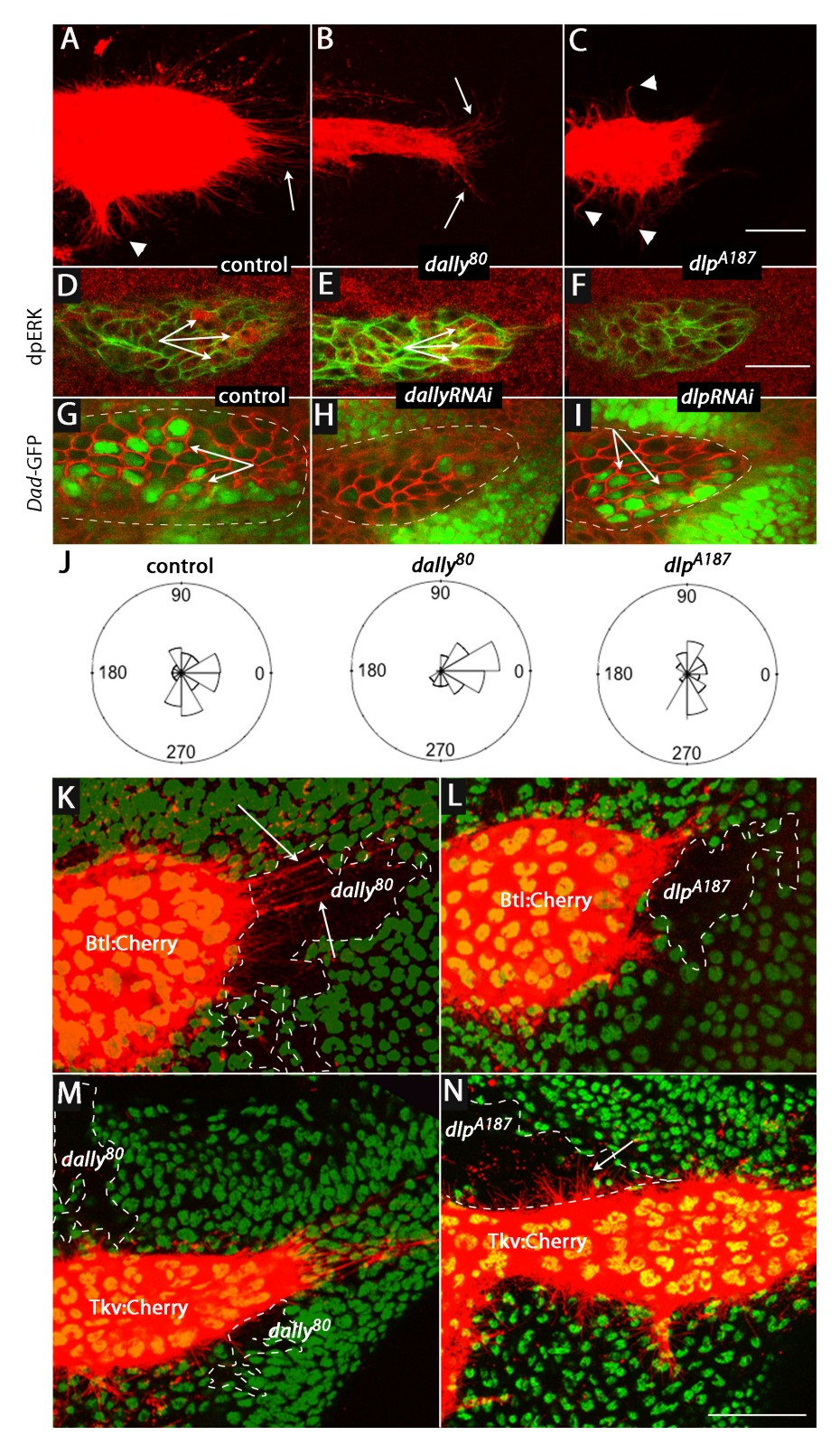

**Figure 6.** ASP cytonemes and ASP signal transduction depend on HSPG expression in underlying disc cells. (A–C) CD8:Cherry expression in the ASP (driven by *btl-Gal4*) marks the ASP and ASP cytonemes in control (A), *dally* (B) and *dlp* (C) mutants. Arrows and arrowheads indicate tip and medial cytonemes, respectively. (D–F) FGF signaling monitored by levels of dpERK staining (arrows) were similar in control (D; *ap-Gal4/+; btl-LHG lexO-CD2: GFP/+*) and *dally* knockdown (E; *ap-Gal4/UAS-dallyRNAi; btl-LHG lexO-CD2:GFP/UAS-dallyRNAi*) ASPs, but was reduced by expression of *dlpRNAi* (F;

*Figure 6 continued on next page*

Figure 6 continued

*ap-Gal4/+; btl-LHG lexO-CD2:GFP/UAS-dlpRNAi*). (G–I) ASPs marked by α-Dlg staining and Dpp signaling monitored by Dad-GFP fluorescence (arrows) in control (G; *ap-Gal4/+; Dad-GFP/+*), *dally* knockdown (H; *ap-Gal4/UAS-dallyRNAi; Dad-GFP/UAS-dallyRNAi*), and *dlp* knockdown (I; *ap-Gal4/+; Dad-GFP/UAS-dlpRNAi*) ASPs. (J) R plots depicting number and orientation of cytonemes in five control and five mutant discs. (K–N) ASP and ASP cytonemes (arrows) were marked by Btl:Cherry (K,L) or by Tkv:Cherry (M,N) and clones of *dally* (K,M) and *dlp* (L,N) are indicated by white dashed lines and absence of GFP. (K) *hs-FLP/+; btl-Gal4 UAS-Btl:Cherry/+; FRT2A GFP/dally^80 FRT2A*; (L) *hs-FLP/+; btl-Gal4 UAS-Btl:Cherry/+; FRT2A GFP/dlp^A187 FRT2A*; (M) *hs-FLP/+; btl-Gal4 UAS-Tkv:Cherry/+; FRT2A GFP/dally^80 FRT2A*; (N) *hs-FLP/+; btl-Gal4 UAS-Tkv:Cherry/+; FRT2A GFP/dlp^A187 FRT2A*. Scale bars: 25 μm.

The following source data is available for figure 6:

**Source data 1.** Figure 6 data - Cytoneme and fluorescence quantification.

the region more proximal to the basal surface of the disc epithelium. *Figure 7F* depicts the distributions of collagen, laminin, Dally and Dlp relative to the location of the Tkv- and Btl-containing cytonemes. The segregation of the Tkv-containing and Btl-containing cytoneme types to separate Dally and Dlp strata and the specific requirements for Dally and Dlp by Tkv-containing and Btl-containing cytonemes establishes the functional importance of ECM stratification.

## Cytoneme-mediated signaling requires integrin function

The Drosophila genome encodes five α integrin subunits (*multiple edematous wings (αPS1, mew), inflated (αPS2, if), scab (αPS3, scb), αPS4* and *αPS5*) and two β integrin subunits (*myospheroid (mys)* and *βv*). We tested for their role in the ASP by expressing RNAi constructs, and observed that whereas the ASP developed normally in the presence of *scbRNAi, αPS4RNAi, αPS5RNAi* and *βvRNAi* (*Figure 8—figure supplement 1A–D*), ASP development was abnormal in the presence of *mysRNAi, mewRNAi* and *ifRNAi* (*Figure 8A–D*). FGF and Dpp signaling in the ASP were reduced in the presence of *mysRNAi, mewRNAi,* or *ifRNAi* (*Figure 8E–L*); and relative numbers of cytonemes were also reduced (*Figure 8M*). Expression of *mysRNAi* and *mewRNAi* had no apparent effect on the number of dividing or apoptotic cells, on cell shape or on the ability of ASP cells to activate Dpp signal transduction (*Figure 8—figure supplement 1E–N*). We also examined control and integrin-depleted preparations to determine whether synaptic contacts between the ASP and wing disc are integrin-dependent. We applied the GRASP (GFP Reconstitution Across Synaptic Partner) technique that generates GFP fluorescence specifically at sites of close and stable cell-cell contacts (*Feinberg et al., 2008*), and that we used previously to mark cytoneme synapses (*Huang and Kornberg, 2015*; *Roy et al., 2014*). As shown in *Figure 8* (panels N-Q), GFP fluorescence was visible at the juxtaposition of the ASP and wing disc in controls, but fluorescence was markedly reduced in preparations from animals that expressed *mysRNAi*. This result is consistent with the reduced numbers of cytonemes in these preparations (*Figure 8B*).

To investigate whether integrin function is also necessary for cytoneme-mediated signaling, we characterized the distribution and role of Integrin-linked kinase (Ilk), a protein that interacts with the cytoplasmic tail of β integrin and links the cytoskeleton and plasma membrane at sites of integrin-mediated adhesion (*Zervas et al., 2001*). We expressed an Ilk:GFP fusion construct in the ASP and observed GFP fluorescence in puncta along and at the tips of ASP cytonemes (*Figure 8R*). Expression of *IlkRNAi* in the ASP perturbed ASP morphogenesis and reduced the number of ASP cytonemes (*Figure 8S*). We conclude that integrin function is required by ASP cytonemes and for signaling in the ASP.

Laminin is an integrin ligand that has been implicated in many developmental and disease contexts, and it is present in the ECM of the wing disc (*Figure 7E,F*) where it is dependent on *pk* function (*Figure 5H,I*). To determine if wing disc laminin is necessary for the ASP, we depleted laminin in the disc by expressing an RNAi construct that targets the α subunit of laminin that is encoded by the *wing blister (wb)* gene. *wb* has been shown to interact genetically with *mys* (*Martin et al., 1999*; *Schöck and Perrimon, 2003*), which encodes a β integrin subunit; the Wb laminin, which has an RGD domain, has been shown to interact with both αPS1 and αPS2 integrins (*Gotwals et al., 1994*; *Graner et al., 1998*). The *wb* knockdown genotype had a shortened and malformed ASP with reduced numbers of cytonemes (*Figures 8M, 9A*), and low levels of Dpp (*Figure 9E*) and FGF

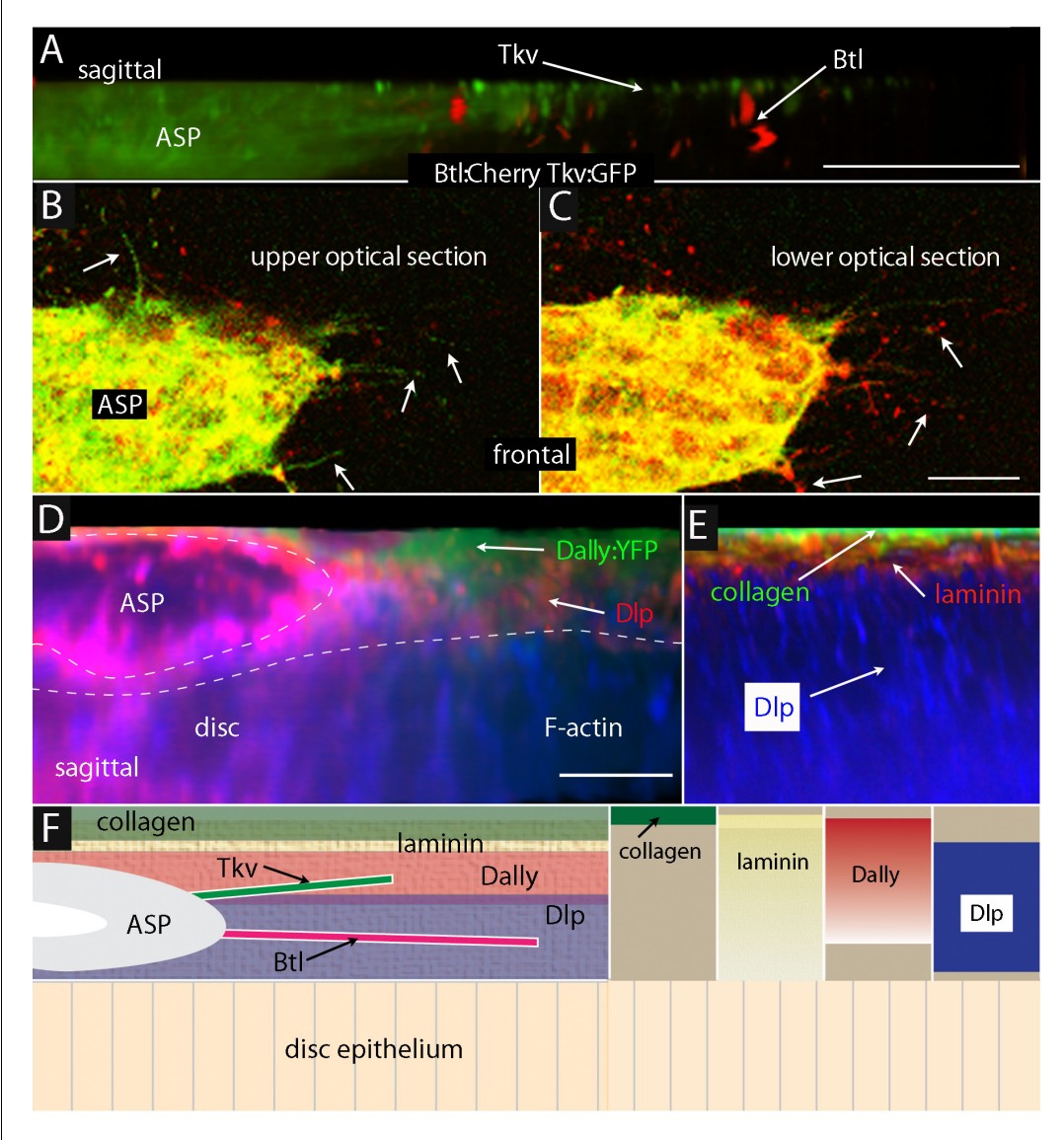

**Figure 7.** ASP cytonemes navigate in a stratified ECM. (**A–C**) Btl:Cherry and Tkv:GFP expressed simultaneously in the ASP (*btl-Gal4 UAS-Btl:Cherry/+; UAS-Tkv:GFP/+*) of a early to mid stage L3. (**A**) Sagittal representation from a composite projection and frontal optical sections (**B,C**) showing ASP cytonemes (arrows) marked by either Btl:Cherry or Tkv:GFP. Tkv:GFP-containing cytonemes lay in focal planes more distant (distal) from the wing disc than cytonemes with Btl:Cherry. (**D**) Sagittal representation from a composite projection showing and early to mid L3 stage wing disc expressing a *dally: YFP* protein trap and stained with α-Dlp antibody (red) and F-actin (blue). The approximate position of ASP indicated by dashed white line; disc is below and not visible in this image. (**E**) Sagittal representation from a merged composite projection showing wing disc expressing a *Viking:GFP* protein trap that marks collagen (green), and stained with antibodies against laminin (red) and Dlp (blue). The approximate position of ASP is indicated by dashed white line. (**F**) Drawings showing our interpretations of the relative locations in an early to mid stage L3 disc of ASP (gray), collagen (green; restricted to the most distal layer), laminin (textured, yellow; most abundant distally and extending proximally to the disc surface), Dally (rose; broadly distributed) and Dlp (lavender; proximal to disc), and Tkv-containing (green) and Btl-containing (red) cytonemes. Scale bars: 20μm.

signaling (*Figure 9I*). Mutants that lack functional *wb* or *mew* are inviable, but *wb/+* and *mew/+* heterozygotes survived and developed normal ASPs that had normal levels of Dpp and FGF signaling (*Figure 9B,C,F,G,J,K*). In contrast, *mew/+; wb/+* double heterozygotes produced abnormal ASPs that had reduced numbers of cytonemes and reduced Dpp and FGF signaling (*Figure 9D,H,L* and *Table 2*). These results suggest that the Wb laminin is a ligand for integrin-dependent cytoneme-ECM interactions.

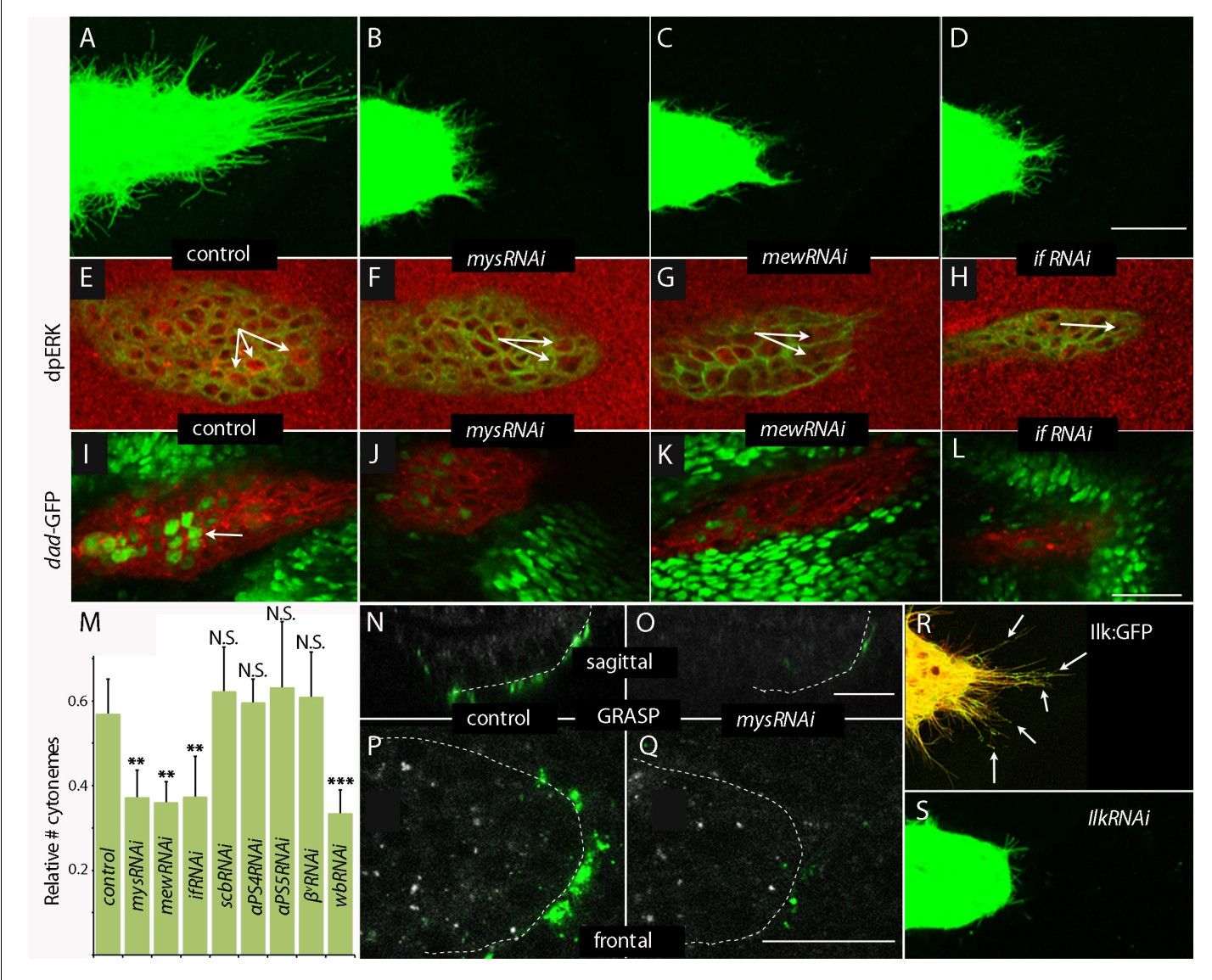

**Figure 8.** Cytoneme-mediated transport requires integrin. (A–D) ASP cytonemes were shorter in ASPs that expressed RNAi against integrin subunits encoded by *mys*, *mew* and *if*. (E–L) ASP cells active in FGF signal transduction were detected with α-dpERK antibody (E) and cells active in Dpp signal transduction were detected by GFP fluorescence (I) in control but not in ASPs depleted of *mys*, *mew* or *if* (F–H, J–L). Arrows point to the ASP cells that express dpERK. Genotypes: (A,E) *btl-Gal4 UAS-CD8:GFP/+*; (B,F) *btl-Gal4 UAS-CD8:GFP/UAS-mysRNAi*; (C,G) *btl-Gal4 UAS-CD8:GFP/UAS-mewRNAi*; (D,H) *btl-Gal4 UAS-CD8:GFP/+; UAS-ifRNAi/+*; (I) *btl-Gal4 UAS-CD8:Cherry/+; dad-GFP/+*; (J) *btl-Gal4 UAS-CD8:Cherry/UAS-mysRNAi; dad-GFP/+*; (K) *btl-Gal4 UAS-CD8:Cherry/UAS-mewRNAi; dad-GFP/+*; (L) *btl-Gal4 UAS-CD8Cherry/+; dad-GFP/UAS-ifRNAi*. (M) Bar graph plots the relative number of cytonemes in ASPs of control and RNAi-mediated integrin-depleted larvae. Error bars: standard deviation; **p<0.01; N.S., not significant. (N–Q) Sagittal and frontal images showing green fluorescence of reconstituted GFP (GRASP) at contacts between ASP cytonemes and Dpp-expressing cells. ASPs outlined by dashed white lines. Genotypes: (N,P) *btl-Gal4 dpp-LHG/+; UAS-CD4:GFP$^{1-10}$ lexO-CD4:GFP$^{11}$/+*; (O,Q) *btl-Gal4 dpp-LHG/UAS-mysRNAi; UAS-CD4:GFP$^{1-10}$ lexO-CD4:GFP$^{11}$/+*. (R) Localization of ILK:GFP (arrows) in ASP cytonemes (*btl-Gal4 UAS-CD8:Cherry/UAS-ILK:GFP*). (S) Number and length of ASP cytonemes was reduced with expression of *ilkRNAi (btl-Gal4 UAS-CD8:GFP/UAS-ilkRNAi)*. Scale bars: 25 μm.

The following source data and figure supplements are available for figure 8:

**Source data 1.** Figure 8 data - Cytoneme and fluorescence quantification.
**Figure supplement 1.** Integrin receptors in the ASP.
**Figure supplement 1—source data 1.** Numbers of ASP cytonemes in *scbRNAi*, *αPS4RNAi*, *βᵛRNAi* and *αPS5RNAi* flies.

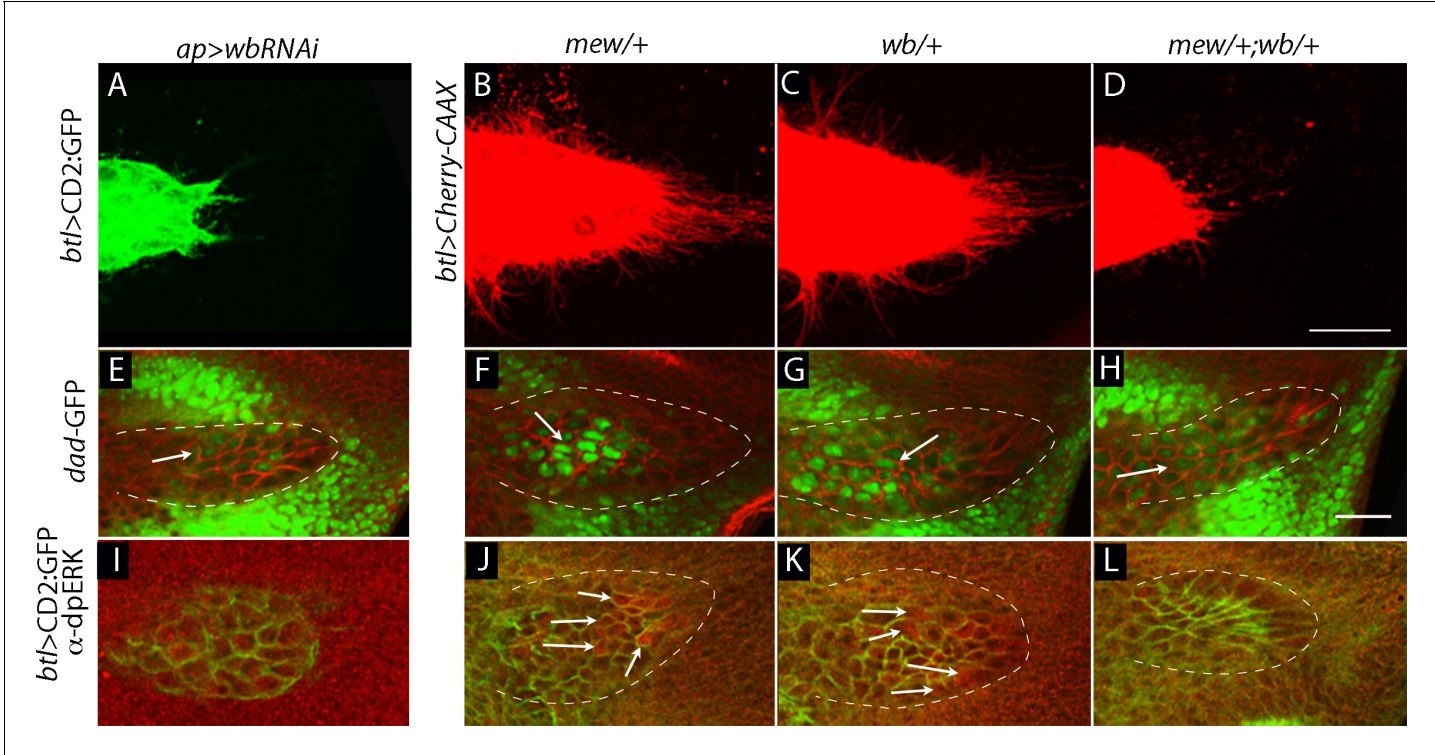

**Figure 9.** Genetic interaction between laminin mutants and integrin mutants. (A,E,I) RNAi targeting expression of the laminin gene *wb* in the wing disc -altered ASP morphology and reduced ASP cytonemes (A) and reduced Dpp (*dad-GFP* fluorescence; E) and FGF signaling (α-dpERK staining; I). Genotypes: (A,I) *ap-Gal4/+; btl-LHG lexO-CD2:GFP/UAS-wbRNAi*; (E) *ap-Gal4/+;dad-GFP/UAS-wbRNAi*. (B–D, F–H, J–L) ASP morphology and number of ASP cytonemes (B,C), Dpp signaling (F,G) and FGF signaling (J,K) was normal in *mew* and *wb* heterozygotes but not in *mew wb* double heterozygotes (D,H,L). Arrows indicate signaling cells and dashed white lines indicate ASPs. Genotypes: (B) *mew[M6]/+; btl-LHG lexO-Cherry-CAAX/+*; (C) *wb[3]/+; btl-LHG lexO-Cherry-CAAX/+*; (D) *mew[M6]/+; wb[3]/+; btl-LHG,lexO-Cherry-CAAX/+*; (F) *mew[M6]/+; dad-GFP/+*; (G) *wb[3]/+; dad-GFP/+*; (H) *mew[M6]/+; wb[3]/+; dad-GFP/+*; (J) *mew[M6]/+*; (K) *wb[3]/+*; (L) *mew[M6]/+; wb[3]/+*. (J–L) Staining was with α-Dlg and α-dpERK antibodies. Scale bar: 25 µm.

The following source data is available for figure 9:

**Source data 1.** Figure 9 data - Cytoneme and fluorescence quantification.

## Discussion

The ASP is a powerful system for studies of cell-cell signaling. Its large cells, easy accessibility for visual analysis, dependence on and sensitivity to paracrine signaling, and robust and abundant cytonemes offer many ways to exploit Drosophila's genetic toolkit. We took advantage of these attributes to address the general question of targeting, asking how ASP cytonemes navigate across the wing disc in order to synapse specifically with cells that express either Dpp or FGF. We focused on the 'intermediate' disc cells that are situated between ASP cells that extend cytonemes and the disc cells they target, and we identified several functions that the intermediate cells must express in order for ASP cytonemes to mediate signaling. The results show that the extracellular environment in which cytonemes navigate is organized and structured and provides essential support that is specific to different cytoneme types.

Cytonemes are dynamic, and their orientations change during development as the tissues that make them grow and the spatial relationship between the cells that produce signaling proteins and the responding cells changes (*Bischoff et al., 2013*; *Roy et al., 2011*). They also change in response to ectopic over-expression of signaling proteins (*Guha et al., 2009*; *Roy et al., 2011*; *Sato and Kornberg, 2002*). These properties suggest that cytonemes are dependent on an active targeting mechanism that conceivably might involve: (1) a random search; (2) chemotaxis informed by an attractant; or (3) structural information encoded in their environment. Although the chemotaxis model seems unlikely because cytonemes that deliver signaling proteins orient to receiving cells

(*Bischoff et al., 2013*; *Callejo et al., 2011*), and although the results we report do not rule out a role for random search, our results indicate that the extracellular environment created by intermediate cells is essential and raise the question how cytonemes interact with it.

We found that the PCP components *pk* and *Vang* are essential for ASP signaling and that ASP cytonemes did not extend over intermediate cells that lack pk or Vang function (*Figure 3*). The strong mutant phenotypes in the ASP had not been noted in previous studies of PCP mutants, the most likely reason being that the dorsal air sacs are not visible in intact flies and flies that lack dorsal air sacs are viable (*Guha et al., 2009*). Because components of the PCP system of the disc has been principally associated with epithelial planar polarity and with polarized subcellular distributions of its core components, the question arises what roles its components might have that affect cytonemes that extend in the extracellular environment.

In other contexts, some PCP components have been implicated in processes that may not be directly involved in epithelial planar polarity - in integrin signaling (*Lewellyn et al., 2013*), microtubule polarity (*Ehaideb et al., 2014*), axon guidance (*Matsubara et al., 2011*; *Shimizu et al., 2011*; reviewed in *Yam and Charron 2013*), polarized extensions of epithelial cells (*Peng et al., 2012*), neural tube closure (*Kibar et al., 2001*), directed cell movements (*Tatin et al., 2013*) and non-canonical Wnt signaling (reviewed in *Karner et al., 2006*). Various mechanisms have been proposed, including integration of cell-matrix and cell-cell interactions (*Dohn et al., 2013*), polarized assembly of fibronectin (*Goto et al., 2005*), and Mmp-dependent remodeling of the ECM (*Williams et al., 2012*). Polarized rows of laminin have been observed in the basement membrane at the basal surface of follicle cells in the Drosophila ovary (*Gutzeit et al., 1991*) that may involve receptor-mediated alignment of the cytoskeleton with the ECM (*Frydman and Spradling, 2001*). Although the functional link between the PCP system and these processes has not been established and although the possibility remains that the roles of the PCP components in these systems are unrelated to their roles in planar polarity, our working assumption is that axon pathfinding, polarized extensions, directed cell migrations and cytoneme navigation may share a requirement for features in the substrates they encounter that are dependent on cell polarity.

We discovered that *pk* and *Vang* mutant cells had reduced amounts of Dally, Dlp and laminin (*Figure 5*). This finding calls into question the extent to which the mutant phenotypes might be due to ECM defects and an indirect consequence of Pk or Vang mis-localization. The related findings that ECM collagen, laminin, and HSPGs are stratified in layers, and that the cytoneme subtypes that contain either Tkv or Btl navigate specifically in the Dally and Dlp ECM layers, respectively (*Figures 6*,*7*), indicates that the ECM is organized, partitioned and regulated to an unanticipated degree and that the ECM might be directly involved in the process of cytoneme extension and pathfinding.

Specific requirements of Dally for Dpp signaling (*Dejima et al., 2011*) and of Dlp for FGF signaling (*Yan and Lin 2007*) have been reported previously and have been interpreted as evidence that Dally and Dlp function as co-receptors; modifications of the Dally and Dlp heparan sulfate backbones have been proposed as structures that provide specificity for the respective signaling systems (*Kamimura et al., 2006* and reviewed in *Nakato and Kimata, 2002*). Our observation that Tkv- and Btl-containing cytoneme subtypes were not present in areas that lacked Dally or Dlp, respectively (*Figure 6*), suggests that each HSPG provides a substrate that supports cytoneme extension and/or stability. This implies that cytonemes interact directly and specifically with the HSPGs and that the roles of these ECM components are not limited to co-receptor functions.

A generally held model posits that axons, migrating cells and cells in developmental fields interpret secreted guidance cues and signaling proteins that encode positional information in the form of chemogradient distributions that are stored in the ECM. HSPGs were proposed to function both to distribute the guidance cues and signaling proteins, and as co-receptors for signaling proteins (*Fujise et al., 2003*; *Yan and Lin, 2007*; *Yan et al., 2010*). Our discovery that cytonemes traffic signaling proteins between cells (*Roy et al., 2014*) revealed that signaling proteins that have been observed between signal producing and receiving cells are cytoneme-associated and are neither extracellular nor ECM-bound (*Roy and Kornberg, 2015*). Our findings are consistent with the idea that secreted signaling proteins and signaling protein receptors are not distributed in the extracellular environment and are not bound to the ECM, but rather that the cytonemes that mediate Dpp and FGF signaling contact the ECM directly in ways that involve both integrins and specific HSPG interactions.

## Materials and methods

### Drosophila stocks

Flies were raised on standard cornmeal and agar medium at 25°C, unless otherwise specified. *bnl*-Gal4/TM6B, *btl*-LHG, *UAS-CD8:GFP*, *UAS-CD8:Cherry*, *lexO-CD2:GFP*, *UAS-Dpp:Cherry*, *UAS-tkv:Cherry*, *UAS-btl:Cherry* and *UAS-btl:GFP* were previously described (*Roy et al., 2014*). *dpp*-LHG and *lexO-mCherry-CAAX*, from K. Basler (*Yagi et al., 2010*); *UAS-ILK:GFP*, from N. Brown (*Zervas et al., 2001*); *UAS-mysRNAi* and *UAS-mewRNAi* (*Han et al., 2012*); *FRT42D Vang$^6$*, from M. Mlodzik (*Wu and Mlodzik, 2008*); *hs-FLP; FRT2A GFP*, from M. Buszczak. *hs-FLP; FRT42D GFP*, from M. Fuller (*Morillo Prado et al., 2012*); *lexO-CD4-GFP$^{11}$* and *UAS-CD4-GFP$^{1–10}$*, from K. Scott; *dpp*-Gal4/CyO and *btl*-Gal4 (*Sato and Kornberg, 2002*); *dad-GFP* (*Ninov et al., 2010*); *UAS-pkRNAi*, from S. Eaton; *dally$^{80}$*, *dlp$^{A187}$* and *UAS-dallyRNAi* (II and III), from H. Nakato (*Akiyama et al., 2008*); *dally-GFP* (Kyoto Stock Center); *UAS-VangRNAi* from the Vienna Drosophila RNAi Center; *pk$^{13}$* (*pk$^{pk-sple13}$*), *Vang$^{153}$* (*Vang$^{stbm-153}$*), *ft$^8$*, *ds$^{UAO71}$*, *ap*-Gal4, *UAS-ifRNAi* and *UAS-dlpRNAi* from the Bloomington Stock Center.

### Sample preparation for live imaging

Wing imaginal discs and their associated trachea were dissected in cold phosphate-buffered saline (PBS), placed on a coverslip with the columnar layer facing the coverslip, and the coverslip was mounted upside-down on a depression slide as previously described (*Huang and Kornberg, 2015*). Images were acquired with an upright Leica TCS SPE confocal microscope using LAS-AF software.

### Clonal analysis

To obtain PCP loss-of-function clones in wing discs: for notum clones, *hs-FLP; FRT42D GFP* females were crossed with *FRT42D pk$^{13}$/CyO; btl-LHG, lexO-mCherry-CAAX/TM6B* males or *FRT42D Vang$^6$/CyO; btl-LHG, lexO-mCherry-CAAX/TM6B* males; for wing blade clones, *hs-FLP; FRT42D-GFP* females were crossed with *pk$^{13}$ FRT42D/CyO* or *Vang$^6$ FRT42D/CyO* males. For MARCM clones, *hs-FLP; tubP-Gal80 FRT40A; tub-Gal4 UAS-GFP/TM6, Tb* females were crossed with *ft$^8$ FRT40A/CyO, act:GFP* males. Progeny were heat-shocked at 38°C for 1 hr between 48 and 72 hr AEL. To generate *dally* or *dlp* mutant clones, *hs-FLP; FRT2A GFP* females were crossed with *btl-Gal4, UAS-btl:Cherry; dally$^{80}$ FRT2A/TM6B, btl-Gal4, UAS-tkv:Cherry; dally$^{80}$ FRT2A/TM6B, btl-Gal4, UAS-btl:Cherry; dlp$^{A187}$ FRT2A/TM6B* males, or *btl-Gal4, UAS-tkv:Cherry; dlp$^{A187}$ FRT2A/TM6B* males. Progeny were heat-shocked at 38°C for 1 hr at the L2 stage (~48 hr AEL), allowed to develop at 25°C, and dissected at the late wandering L3 stage.

### Immunohistochemistry

L3 larvae were dissected in cold PBS and wing discs together with Tr2 trachea were fixed in 4% formaldehyde. After extensive washing, the samples were permeablized with TritonX-100 and then blocked in 10% donkey serum, and incubated with primary antibodies that had been diluted in blocking buffer. The following primary antibodies were used: α-dpERK (Sigma); α-laminin (from J. Fessler; *Fessler et al., 1987*); α-pMad (from E. Laufer and P. ten Dijke) (*Persson et al., 1998*); α-Senseless (from H. Bellen), α-Discs large and α-Dally-like (Developmental Studies Hybridoma Bank); α-cleaved Caspase-3 (Asp$^{175}$) and α-phosphohistone H3 (Ser$^{10}$) (Cell Signaling Technology, Danvers, MA). Secondary antibodies were conjugated to Alexa Fluor 405, 488, 555, or 647. Samples were mounted in Vectashield.

### Image quantification and statistical analysis

ASP cytonemes were counted and measured in z-section stacks of images from five ASPs for each genotype. Lengths represent distance from each tip along the connecting shaft to the widening base at the plasma membrane. The ratios represent the mean value for number of cytonemes per unit length along the circumference of the ASP together with standard deviation. R plots were made for five preparations of each genotype and rose diagrams were generated by R software. For intensity measurements using Image J, a rectangle in the wing disc or ASP that included approximately ten cells was chosen. The average values (with background fluorescence subtracted) are presented.

## qRT-PCR

Total RNA was extracted from wing discs of 30 larvae using with the Zymo Research RNA MicroPrep (Cat. #R1060). Reverse transcription was carried out using the Applied Biosystem High Capacity RNA-to-cDNA (Cat. #4387406). qPCR reactions were performed with a BioRad C1000 Touch Thermal Cycler and SYBR Green (Bioline). qPCR results were analyzed according to the comparative threshold cycle (Ct) method, where the amount of target, normalized to an endogenous *actin* reference and relative to an experimental control, is given by $2^{-\triangle\triangle Ct}$. Ct represents the PCR cycle number at which the amount of target reaches a fixed threshold. The $\triangle$Ct value is determined by subtracting the reference Ct value (i.e. actin) from the target Ct value. $\triangle\triangle$Ct was calculated by subtracting the $\triangle$Ct experimental control value.

## Acknowledgements

We thank K Basler, N Brown, C Han, M Mlodzik, M Buszczak, M Fuller, K Scott, S Eaton, H Nakato, E Martín-Blanco, J Axelrod, M Affolter, R Palmer, C Zervas, the Vienna Drosophila RNAi Center, Kyoto Stock Center, and Bloomington Stock Center for fly stocks; E Laufer, P ten Dijke, H Bellen, C Desplan and the Developmental Studies Hybridoma Bank for antibodies; S Roy for his advice and help; D Sheppard, I Guerrero, P Lawrence, S Eaton, J Axelrod, P Adler, J Esko for suggestions on the manuscript, and all members of the Kornberg lab for discussions and constructive suggestions.

## Additional information

### Funding

| Funder | Grant reference number | Author |
|---|---|---|
| National Institutes of Health | 5T32HL007731 | Hai Huang |
| National Institutes of Health | GM030637 | Thomas B Kornberg |

The funders had no role in study design, data collection and interpretation, or the decision to submit the work for publication.

### Author contributions

HH, Conception and design, Acquisition of data, Analysis and interpretation of data, Drafting or revising the article; TBK, Conception and design, Analysis and interpretation of data

### Author ORCIDs

Thomas B Kornberg, http://orcid.org/0000-0002-6879-7066

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
