## [Decision Letter]

Thank you for submitting your article "Cells must express planar cell polarity functions and extracellular matrix components to support cytonemes" for consideration by *eLife*. Your article has been reviewed by three peer reviewers, and the evaluation has been overseen by Jeremy Nathans as the Reviewing Editor and Fiona Watt as the Senior Editor. The reviewers have opted to remain anonymous.

Your revised *eLife* manuscript has been re-reviewed by the same three expert reviewers who conducted the initial review and the Reviewing Editor has drafted this decision to help you prepare a revised submission.

The reviewers have a series of specific comments that are detailed below. In our discussions, the consensus view of the reviewers and of this editor can be summarized as follows:

1) Figure 8 and Figure 9 provide evidence that integrins and laminin are required for cytonemes, but this appears tangential because of the excessive emphasis that the authors put on PCP. Indeed, PCP is barely mentioned in the Abstract despite its top billing in the title. A better title would be "Identification of genes required for cytoneme localization and cytoneme-dependent signaling". At the very least, omit the word "must" from the title.

2) The interpretations should be clear, and while the experiments have gone some way to clarifying whether the effects of Vang/Pk are due to planar polarity, there is still some ambiguity in the text, and the authors have not tried very hard to address this issue with new experiments.

3) The authors need to substantively address the reviewers' and editor's comments.

We look forward to receiving your revised manuscript. Given that this would in essence be the second opportunity for revision of this work, we must make a binding and final decision on the next version.

*Reviewer #1:*

The paper is more focused now, and I feel that the balance between provocative findings and mechanistic exploration is good. Several points should still be addressed in the text:

1) It appears that while Pk and Vang are involved, this is not the classical PCP pathway. Not only because RNAi for other PCP genes did not give a phenotype, but also because overexpression of the other PCP genes had not effect. This point was also raised by the other two reviewers, and is not sufficiently clarified in the text.

2) How to stratify GPI anchored HSPGs is an intriguing question which should be raised in the Discussion.

3) While the relationship to integrins and laminins is highlighted as the main finding, I found this part less compelling, and think that the paper would be tighter if Figure 8–Figure 9 would not be included.

4) Additional questions to pose in the Discussion:

How do cytonemes sense different ECM components?

If the ECM provides only a permissive environment (I loved the "go west boy" analogy), what provides that actual guidance for cytonemes?

*Reviewer #2:*

My original impression of the first draft of this manuscript was that it contained two sets of observations (the effects of the PCP proteins on cytoneme navigation, and the stratification of the ECM), neither of which was very well developed. I suggested that the authors revise the manuscript one of two ways.

First was to test more rigorously whether PCP was encoded in the ECM. In particular, it was not clear whether the PCP proteins had an instructive role (e.g. by polarizing the ECM). It was also not clear whether all the PCP proteins were involved in the process they describe, or only Vang and Pk (suggesting a non-PCP role for these proteins). The authors now show overexpression experiments in a supplemental figure, which find that overexpression of Vang and Pk affect cytoneme number, but overexpression of Fmi and Fz do not – suggesting that the effect they report may be specific to Vang and Pk. The authors also have made some textual changes, particularly in the Discussion, to reflect my concerns that the effects of Vang and Pk on cytoneme navigation may not be due to their roles in mediating planar polarity pathway activity. Overall, their data shows that Vang and Pk affect the levels of Dlp, Dally and laminin post-transcriptionally. The mechanism by which they do this is not explored, and therefore this is of limited interest.

Alternatively, I suggested that the authors focus more on their most interesting observation: that the ECM is stratified and that Dpp-receiving and FGF-receiving cytonemes navigate differentially in the Dally and Dlp layers. However, in this revised version the authors have not explored the mechanistic basis for this any further.

*Reviewer #3:*

This manuscript investigates genetic requirements for the formation of cytonemes and cytoneme-mediated signaling to the dorsal air sac primordial. The authors identify requirements for pk and vang, which correlate with reduced levels of laminin, and the glypicans Dally and Dlp. They then investigate requirements for Dlp and Dally, and report that Dpp signaling, and Tkv-labeled cytonemes, require Dally, whereas FGF signaling, and Btl-labeled cytonemes, require Dlp. They then claim that the ECM is stratified, with Dally and Dlp in different ECM layers, and that Tkv and Btl cytonemes form preferentially in these different layers. Finally, they report that integrins and laminin are also required for cytonemes, and cytoneme-dependent signaling.

The manuscript contains several novel and exciting observations that influence our understanding of intercellular signaling, and thus I think in principle it should be suitable for publication in *eLife*. However, there remain flaws in the presentation, and the authors have failed to address some key issues that were raised previously.

One major concern, raised by all of the reviewers of the previous version, yet still not addressed, are the authors' mis-leading statements throughout the text claiming a requirement for "planar cell polarity functions". The authors identify a requirement for pk and vang, but see no effect of knockdown of other genes in the Fz-Vang PCP pathway, which clearly argues against a role for this PCP pathway. Instead, it appears that they have identified a requirement for Pk and Vang that is distinct form their function in PCP. In its present form, I would have to say that the text makes claims that are not supported by the data, so the manuscript is not suitable for publication. They report an influence of Fat and Ds, but this is not characterized well enough to provide any additional support for the claim of a requirements for a "PCP system" e.g. – is it due to effects of Fat and Ds on Pk localization? – Is it due to effects of Fat and Ds on Hippo signaling? If they revised the text to replace "PCP system" and "PCP function" with "pk and vang" this problem would be solved.

A second concern, also raised previously, is that the authors make claims about stratification of Dally and Dlp localization in the ECM, without actually providing any data showing that the Dlp observed is extracellular. The authors' response here claiming that it would be "technically challenging" is not compelling: the experiments required are feasible, and they need to do them. It's a significant issue because an alternative interpretation of Figure 7 could be that extracellular Dlp is too low to be detected, so what they are claiming as stratification could be extracellular Dally over cytoplasmic Dlp.

Additional Points:

A claim is made in the text that "Laminin was also most prominent distally but it did not extend as far from the disc surface as collagen." I can't see this in the cited figure panel (7E).

In the Discussion, the authors state "In other contexts, some PCP components have been implicated in processes that appear to have little in common with planar polarity" They then list several functions that clearly involve planar polarization of cells, including axon pathfinding, polarized extensions, directed cell migrations.

In Figure 7 the authors present data showing that Btl and Tkv cytonemes extending from the tip of the ASP are in different focal planes, leading them to infer that they navigate through different ECM layers. However, elsewhere in the manuscript they emphasize that Tkv cytonemes extend from the lateral sides of the ASP, and Btl cytonemes form the tip of the ASP, so these Tkv tip cytonemes may be unusual. Couldn't an alternative explanation be that these Tkv cytonemes at the tip are simply lost and non-functional? Can we see the same thing looking at lateral cytonemes?

Figure 7—figure supplement 1. Multiple panels are shown with no explanation of how they differ from each other and what we are supposed to conclude from this set of images.

[Editors’ note: a previous version of this study was rejected after peer review, but the authors submitted for reconsideration. The previous decision letter after peer review is shown below.]

Thank you for submitting your work entitled "Cytoneme navigation requires the planar cell polarity system and specific components of the extracellular matrix" for consideration by *eLife*. Your article has been reviewed by three peer reviewers, and the evaluation has been overseen by a Reviewing Editor and Fiona Watt as the Senior Editor. Our decision has been reached after consultation between the reviewers. Based on these discussions and the individual reviews below, we regret to inform you that your work will not be considered further for publication in *eLife*.

While the reviewers found the work interesting, the number of substantive questions raised was such that we feel we must reject it. We hope that the reviewers' comments will be useful to you. We apologize for not being able to deliver better news, and we hope that you will continue to consider *eLife* for future submissions. The reviewers' comments are appended.

*Reviewer #1:*

The paper by Huang and Kornberg utilizes the larval air sac premordium system, developed by the Kornberg lab, in order to gain insights into the signals that may direct cytonemes to their ultimate targets, namely the cells producing Dpp and Branchless, respectively. They find that several planar polarity elements are required, that different HSPGs are required for the two types of cytonemes, and that these HSPGs are segregated into distinct layers. Finally, the involvement of some integrins and their ligands is also implicated.

The previous groundbreaking work of the Kornberg lab has brought the ASP system to a level where it is possible to assay, at a high resolution, the migration of the two cytoneme types. Based on this system, the reported results are convincing. Clearly, the guidance mechanisms for cytonemes that are coordinated with the cells they encounter along the way (other than the ligand-producing cells) are of general interest and importance.

1) This paper has shown that some aspects of the planar polarity system may participate, and their level of expression may be important. How a system that generates local asymmetry between adjacent cell contacts may contribute to long-range migration cues is not clear.

2) The paper has shown that the HSPGs Dally and Dlp are stratified and are specifically required for each of the two types of cytonemes. The mechanism for stratification is not clear, since the Dally protein that is further away from the disc epithelium may need to be released from the apical membrane of the disc epithelium. The migration of each branch type appears to be restricted or focused at a distinct HSPG layer. How each HSPG contributes to directed migration is not known.

3) Finally, some integrins and their ligands are also important for cytoneme migration.

Every paper has a fine balance between the open questions that are presented and resolved, and the new and intriguing questions that are opened up. My main problem with this paper is that it leans too heavily towards opening new questions, without satisfactorily resolving at a mechanistic level any of the original questions it posed. All of these open questions are superimposed on the fact that the basis for segregation of BMP receptors vs. FGF receptors (and perhaps other membrane proteins that have not been identified) to the two cytoneme types is not known. Thus, the differential response of the cytonemes to external cues, other than BMP and FGF, may rely on unknown differences in their composition. The manuscript could serve as a basis for several excellent papers in the future, once the mechanistic aspects behind each of the findings are better resolved.

*Reviewer #2:*

The title of this manuscript says it is about cytoneme navigation requiring the planar polarity system and interactions with the ECM. In the Abstract they also suggest that cytoneme extension may be guided by PCP-dependent polarity encoded in or dependent on the ECM. This led me to suppose that they had shown that the core PCP system somehow laid down polarized ECM (comparable to the polarized fibronectin deposits controlled by PCP genes in *Xenopus*). However, this was misleading, as the link between the PCP system and the ECM is left unclear. The authors show that some PCP genes (Vang and pk) affect cytoneme function, but ultimately this appears to be due to a decrease in overall Dally and Dlp levels. They do not investigate this further, and no polarization of the ECM is shown. Thus the only evidence is for a permissive rather than an instructive role for Vang and Pk.

The middle section of the Results is the most interesting. The authors make the interesting observation that the ECM on the basal surface of the wing disc is stratified into collagen, laminin, Dlp and Dally layers. Cytonemes extending from the air sac primordium (ASP) are known to respond to both Dpp and FGF (produced in specific regions of the wing disc). Cytonemes responding to Dpp navigate in the Dally layer, and fail to extend into dally mutant tissue, whilst cytonemes responding to FGF localize in the Dlp layer, and fail to extend into Dlp mutant tissue.

Prior to this, the authors devote four figures to showing how Vang and pk promote cytoneme extension. However, in Figure 5 they reveal that levels of Dally and Dlp (and less convincingly, laminin) are reduced in Vang and pk mutants, and this may be the reason for the defect in cytoneme navigation.

The final section investigates further components of the ECM that could affect cytoneme navigation, but these do not tie in very well with the Dally/Dlp story. They show that integrin signaling is seen in the tips of cytonemes produced by the ASP, and that both integrins (in the ASP) and their ligand laminin (expressed in the wing disc) are required for cytoneme formation. It is not clear whether these cytonemes are different to those responding to Dpp or FGF, as they are in different layers of the ECM.

In summary, as it stands I do not recommend publication. I think there are two options. One would be to redraft and resubmit a new manuscript with a focus on the ECM stratification. The role of the PCP system in regulating ECM production is of interest, but the authors should avoid the implication in the title and the abstract that PCP systems either polarize the ECM or interact more specifically with cytoneme navigation. Further experiments would be necessary, perhaps exploring in more detail the mechanisms by which the ECM affects cytoneme navigation, or alternatively, how does ECM stratification occur? Do different components of the ECM interact to sort into layers, and does loss of one component affect the overall structure?

For the second option, the authors could do more experiments to rigorously test the hypothesis that PCP is encoded in the ECM. For example, the authors could determine if cytoneme extension correlates with polarity. What happens to cytonemes if Vang or Pk are overexpressed (and thus they are no longer polarized)? Do Vang/Pk provide instructive directional cues or are they merely permissive?

In addition, to substantiate the claim that the PCP system is involved in cytoneme navigation (albeit indirectly via the ECM) it would be necessary to investigate this further. Currently the authors only examine the effects of two core PCP mutants (Vang and pk) on cytoneme morphology. No cytoneme defects are seen using fmi, fz and dsh RNAi, and the authors do not look at mutants. This leads me to wonder if the effects they see do actually involve the core PCP system, or if they involve another non-polarity role for Vang and Pk. It would be essential to determine that fmi, fz or dsh (using the PCP-specific dsh[1] allele) mutant clones do in fact have the same phenotype as Vang and pk mutants.

*Reviewer #3:*

This manuscript describes an investigation of genetic requirements for the formation of cytonemes from the air sac primodia (ASP) of the *Drosophila* wing imaginal disc. The authors report that extension of these cytonemes is severely impaired if disc cells are lacking the planar cell polarity (PCP) genes pk or vang. They provide evidence that this is due to reduced levels of the glypicans Dally and Dlp on the basal side of these cells. Moreover, they provide evidence that Dally contributes to the formation of Tkv-containing cytonemes, and hence Dpp signaling, whereas Dlp contributes to formation of Btl-containing cytonemes, and hence FGF signaling. They correlate this with differential localization of these glypicans, and dependent cytonemes, within the extracellular matrix (ecm). They also identify requirements for integrin signaling in cytoneme formation or stability. Their observations are novel and very interesting, and for the most part well supported, but I do have concerns with some of the data.

1) Some of the reported effects on expression levels are rather subtle, yet there is no information provided as to how changes in expression levels were assessed – no quantitation or scoring criteria, no indication of numbers scored.e.g. Figure 5, Figure 6, Figure 8, Figure 9, Figure 4—figure supplement 1A-H.

2) The claim that Tkv and Btl-containing cytonemes form in different layers is intriguing, but supported only by 3 figure panels (Figure 7), with no indication of numbers scored or quantitation. Also, the authors infer that this reflects formation or migration in Dlp vs Dally containing ecm layers, but couldn't this be tested experimentally, e.g. by double staining for Dally or Dlp and Tkv or Btl?

3) The authors claim (Figure 7) the ecm near the ASP is 20 μm thick, which I find surprising. Despite drawing a dashed line for the basal surface of the disc epithelium, they don't report using any markers for the basal surface of these cells (e.g. they could use F-actin, or integrins), and I think this is needed to clarify the localizations of Dally and Dlp (e.g., is all the Dlp detected really all extracellular?).

4) The title "cytoneme navigation requires the planar polarity system", and the interpretation in the text that goes along with it, is over-stated. The authors clearly show a requirement for Pk. But Pk has also been reported to have non-PCP functions. Most notably, it’s been implicated in microtubule orientation, and vesicle transport along microtubules unrelated to PCP (e.g. Ehaideb et al. 2014), which seems more likely to account for the reduced basal secretion of glypicans than the core PCP system, whose components localize to the apical side of epithelial cells. The authors also report an effect of Vang, but if I understand Table 1 correctly, fz, dsh, and flamingo had no effect, which would be inconsistent with a role for the core PCP system. Perhaps the influence of Vang could be accounted for by an influence on the levels or localization of Pk? In any event, I think the authors should give rather more weight to the likely-hood that the requirements for Pk and Vang do not reflect a role for PCP here.

---

## [Author Response]

*The reviewers have a series of specific comments that are detailed below. In our discussions, the consensus view of the reviewers and of this editor can be summarized as follows:*

We thank the reviewers and editor for their suggestions and comments and have revised the manuscript as follows.

*1) Figure 8 and Figure 9 provide evidence that integrins and laminin are required for cytonemes, but this appears tangential because of the excessive emphasis that the authors put on PCP. Indeed, PCP is barely mentioned in the Abstract despite its top billing in the title. A better title would be "Identification of genes required for cytoneme localization and cytoneme-dependent signaling". At the very least, omit the word "must" from the title.*

We have changed the title, but have not adopted the reviewer/editor suggestion for several reasons. First, the suggested title is general and does not make the main point of the paper – that the functions we identified *must* be expressed by cells *in the disc* for cytonemes to extend from the ASP and for cytoneme-mediated signaling. Second, we refer to PCP and ECM in the title as the best way to summarize the many genes (not only pk and Vang) that we identified.

We articulate the key points in the Abstract as best we can within the allotted space; we do not understand the comment “barely mention”.

*2) The interpretations should be clear, and while the experiments have gone some way to clarifying whether the effects of Vang/Pk are due to planar polarity, there is still some ambiguity in the text, and the authors have not tried very hard to address this issue with new experiments.*

*3) The authors need to substantively address the reviewers' and editor's comments.*

*We look forward to receiving your revised manuscript. Given that this would in essence be the second opportunity for revision of this work, we must make a binding and final decision on the next version.*

Reviewer #1:

*The paper is more focused now, and I feel that the balance between provocative findings and mechanistic exploration is good. Several points should still be addressed in the text:*

*1) It appears that while Pk and Vang are involved, this is not the classical PCP pathway. Not only because RNAi for other PCP genes did not give a phenotype, but also because overexpression of the other PCP genes had not effect. This point was also raised by the other two reviewers, and is not sufficiently clarified in the text.*

*2) How to stratify GPI anchored HSPGs is an intriguing question which should be raised in the Discussion.*

We did not expand the Discussion to topics about which we cannot do more than speculate (how the HSPGs might be stratified, how cytonemes might sense the HSPGs, how the ECM might guide cytonemes).

*3) While the relationship to integrins and laminins is highlighted as the main finding, I found this part less compelling, and think that the paper would be tighter if Figure 8–Figure 9 would not be included.*

We did not remove Figure 8 and Figure 9 because they establish the importance of integrin signaling to the interaction between cytonemes and the ECM, which is relevant to the questions this work addresses.

*4) Additional questions to pose in the Discussion:*

*How do cytonemes sense different ECM components?*

*If the ECM provides only a permissive environment (I loved the "go west boy" analogy), what provides that actual guidance for cytonemes?*

*Reviewer #2:*

*My original impression of the first draft of this manuscript was that it contained two sets of observations (the effects of the PCP proteins on cytoneme navigation, and the stratification of the ECM), neither of which was very well developed. I suggested that the authors revise the manuscript one of two ways.*

*First was to test more rigorously whether PCP was encoded in the ECM. In particular, it was not clear whether the PCP proteins had an instructive role (e.g. by polarizing the ECM). It was also not clear whether all the PCP proteins were involved in the process they describe, or only Vang and Pk (suggesting a non-PCP role for these proteins). The authors now show overexpression experiments in a supplemental figure, which find that overexpression of Vang and Pk affect cytoneme number, but overexpression of Fmi and Fz do not – suggesting that the effect they report may be specific to Vang and Pk. The authors also have made some textual changes, particularly in the Discussion, to reflect my concerns that the effects of Vang and Pk on cytoneme navigation may not be due to their roles in mediating planar polarity pathway activity. Overall, their data shows that Vang and Pk affect the levels of Dlp, Dally and laminin post-transcriptionally. The mechanism by which they do this is not explored, and therefore this is of limited interest.*

We try to more precisely refer to the genes we identified as components of the PCP system rather than PCP functions (apparently the more common practice) and to other contexts where the PCP components are involved.

*Alternatively, I suggested that the authors focus more on their most interesting observation: that the ECM is stratified and that Dpp-receiving and FGF-receiving cytonemes navigate differentially in the Dally and Dlp layers. However, in this revised version the authors have not explored the mechanistic basis for this any further.*

Reviewer #3:

*This manuscript investigates genetic requirements for the formation of cytonemes and cytoneme-mediated signaling to the dorsal air sac primordial. The authors identify requirements for pk and vang, which correlate with reduced levels of laminin, and the glypicans Dally and Dlp. They then investigate requirements for Dlp and Dally, and report that Dpp signaling, and Tkv-labeled cytonemes, require Dally, whereas FGF signaling, and Btl-labeled cytonemes, require Dlp. They then claim that the ECM is stratified, with Dally and Dlp in different ECM layers, and that Tkv and Btl cytonemes form preferentially in these different layers. Finally, they report that integrins and laminin are also required for cytonemes, and cytoneme-dependent signaling.*

*The manuscript contains several novel and exciting observations that influence our understanding of intercellular signaling, and thus I think in principle it should be suitable for publication in* eLife*. However, there remain flaws in the presentation, and the authors have failed to address some key issues that were raised previously.*

*One major concern, raised by all of the reviewers of the previous version, yet still not addressed, are the authors' mis-leading statements throughout the text claiming a requirement for "planar cell polarity functions". The authors identify a requirement for pk and vang, but see no effect of knockdown of other genes in the Fz-Vang PCP pathway, which clearly argues against a role for this PCP pathway. Instead, it appears that they have identified a requirement for Pk and Vang that is distinct form their function in PCP. In its present form, I would have to say that the text makes claims that are not supported by the data, so the manuscript is not suitable for publication. They report an influence of Fat and Ds, but this is not characterized well enough to provide any additional support for the claim of a requirements for a "PCP system" e.g. – is it due to effects of Fat and Ds on Pk localization? Is it due to effects of Fat and Ds on Hippo signaling? If they revised the text to replace "PCP system" and "PCP function" with "pk and vang" this problem would be solved.*

*A second concern, also raised previously, is that the authors make claims about stratification of Dally and Dlp localization in the ECM, without actually providing any data showing that the Dlp observed is extracellular. The authors' response here claiming that it would be "technically challenging" is not compelling: the experiments required are feasible, and they need to do them. It's a significant issue because an alternative interpretation of Figure 7 could be that extracellular Dlp is too low to be detected, so what they are claiming as stratification could be extracellular Dally over cytoplasmic Dlp.*

Figure 7 has been modified to include better images that place Dally and Dlp relative to the disc and the ASP and that distinguish between the location of laminin and collagen.

Additional Points:

*A claim is made in the text that "Laminin was also most prominent distally but it did not extend as far from the disc surface as collagen." I can't see this in the cited figure panel (7E).*

*In the Discussion, the authors state "In other contexts, some PCP components have been implicated in processes that appear to have little in common with planar polarity" They then list several functions that clearly involve planar polarization of cells, including axon pathfinding, polarized extensions, directed cell migrations.*

*In Figure 7 the authors present data showing that Btl and Tkv cytonemes extending from the tip of the ASP are in different focal planes, leading them to infer that they navigate through different ECM layers. However, elsewhere in the manuscript they emphasize that Tkv cytonemes extend from the lateral sides of the ASP, and Btl cytonemes form the tip of the ASP, so these Tkv tip cytonemes may be unusual. Couldn't an alternative explanation be that these Tkv cytonemes at the tip are simply lost and non-functional? Can we see the same thing looking at lateral cytonemes?*

The issues raised by Reviewer #3 regarding the differences between tip and lateral cytonemes is a function of developmental stage (both Tkv- and Btl-containing cytonemes extend from the tip of the ASP at early to mid stage L3, but at late L3 Tkv-containing cytonemes extend from the medial region of the ASP and Btl-containing cytonemes extend from the tip. These details have been added to the text and to the Figure legend.

Figure 7—figure supplement 1*. Multiple panels are shown with no explanation of how they differ from each other and what we are supposed to conclude from this set of images.*

Figure 7—figure supplement 1 has been deleted.

[Editors’ note: the author responses to the previous round of peer review follow.]

Reviewer #1:

*[…] Every paper has a fine balance between the open questions that are presented and resolved, and the new and intriguing questions that are opened up. My main problem with this paper is that it leans too heavily towards opening new questions, without satisfactorily resolving at a mechanistic level any of the original questions it posed. All of these open questions are superimposed on the fact that the basis for segregation of BMP receptors vs. FGF receptors (and perhaps other membrane proteins that have not been identified) to the two cytoneme types is not known. Thus, the differential response of the cytonemes to external cues, other than BMP and FGF, may rely on unknown differences in their composition. The manuscript could serve as a basis for several excellent papers in the future, once the mechanistic aspects behind each of the findings are better resolved.*

We integrate our findings with the states of the fields of proteoglycan biology, ECM structure and function, and the PCP system, a task made especially challenging because there were no good precedents for any of the major findings we made – that several components of the PCP system play a major role in determining the composition of the basal ECM, that the components of the basal ECM are roughly stratified into layers above the surface of the disc epithelium, that the stratification of the ECM is functionally important, and that different cytoneme types require specific HSPGs and navigate in specific ECM layers. In our opinion, these are significant, “ground breaking” findings, so we clearly failed to describe them appropriately in our manuscript. We thank the reviewers for their suggested changes that might improve it. In the revision we are now submitting, we tried to fix the core problematic issues that they identified.

The reviewer points out that “every paper has a fine balance between the open questions that are presented and resolved, and the new and intriguing questions that are opened up” and concludes that the work lacks insight into mechanism. With all due respect, we disagree with this judgment. We prefer to re-phrase the proposition as “every paper should report solid and well-supported findings that improve and change our perception of the underlying biology”. That is, the balance is between “pedantic” description and novel insight, and if the work does not offer new ways to think about an interesting biological question, even well-supported descriptions amount to no more than isolated facts and lack general interest, or in today’s jargon, “do not rise to the mechanistic level”. The findings we report are very well supported by the data, and they offer several major, ground-breaking and potentially “paradigm shifting” findings, which by their nature open up many fascinating questions. Although we agree with the reviewer that “this manuscript could serve as a basis for several excellent papers in the future, once the mechanistic aspects behind each of the findings are better resolved”, these particular questions and directions do not fit with our immediate interests. We hope that this manuscript would be judged for the quantity and quality of its findings and not for the number of questions that remain.

Reviewer #2:

*The title of this manuscript says it is about cytoneme navigation requiring the planar polarity system and interactions with the ECM. In the Abstract they also suggest that cytoneme extension may be guided by PCP-dependent polarity encoded in or dependent on the ECM. This led me to suppose that they had shown that the core PCP system somehow laid down polarized ECM (comparable to the polarized fibronectin deposits controlled by PCP genes in Xenopus). However, this was misleading, as the link between the PCP system and the ECM is left unclear. The authors show that some PCP genes (Vang and pk) affect cytoneme function, but ultimately this appears to be due to a decrease in overall Dally and Dlp levels. They do not investigate this further, and no polarization of the ECM is shown. Thus the only evidence is for a permissive rather than an instructive role for Vang and Pk.*

We agree with the reviewer that the role of the PCP genes in cytoneme-mediated signaling should not be linked to the role of these genes in planar polarity. We also agree that absence of cytoneme extensions in the various mutant conditions we analyzed does not directly implicate a process for path choice. These are excellent points and we have revised the manuscript accordingly. Our results suggest that functional ECM for cytoneme navigation requires PCP genes, pk and Vang. The title and Abstract have been revised. We agree that it would be interesting to show polarization of the ECM, but we have not observed apparent polarized patterns of ECM components using methods and reagents that are currently available.

*The middle section of the Results is the most interesting. […] I think there are two options. One would be to redraft and resubmit a new manuscript with a focus on the ECM stratification. The role of the PCP system in regulating ECM production is of interest, but the authors should avoid the implication in the title and the abstract that PCP systems either polarize the ECM or interact more specifically with cytoneme navigation. Further experiments would be necessary, perhaps exploring in more detail the mechanisms by which the ECM affects cytoneme navigation, or alternatively, how does ECM stratification occur? Do different components of the ECM interact to sort into layers, and does loss of one component affect the overall structure?*

The reviewer offers an option to redraft the manuscript and states that “further experiments would be necessary, perhaps exploring in more detail the mechanisms by which the ECM affects cytoneme navigation, or alternatively, how does ECM stratification occur? Do different components of the ECM interact to sort into layers, and does loss of one component affect the overall structure?” Certainly these are fascinating and interesting questions, but we do not agree that answering them is necessary to establish the importance and significance of our discovery of the functional stratification of the ECM. This finding is well supported by the data we present and we do not understand why the relative number of figure panels is germane. Although we agree that the suggested questions are interesting and worthy of pursuit, we have attempted without success to use available reagents to visualize oriented structures in the ECM, and it is not clear that the methods and reagents that would be needed to provide definitive answers are currently available. The reviewer did not suggest specific experiments to tackle the general questions that we agree are of interest, and it is beyond the scope of this study and peripheral to our immediate goals to undertake a long-term study of the cell biology responsible for cytoneme specificity or for the structure and function of the ECM that answering them would require.

*For the second option, the authors could do more experiments to rigorously test the hypothesis that PCP is encoded in the ECM. For example, the authors could determine if cytoneme extension correlates with polarity. What happens to cytonemes if Vang or Pk are overexpressed (and thus they are no longer polarized)? Do Vang/Pk provide instructive directional cues or are they merely permissive?*

We performed over-expression tests for Vang and Pk. The results revealed that over-expression of *Vang* or *pk* in the disc decreased the number of ASP cytonemes. We added the results in Figure 2—figure supplement 1.

*In addition, to substantiate the claim that the PCP system is involved in cytoneme navigation (albeit indirectly via the ECM) it would be necessary to investigate this further. Currently the authors only examine the effects of two core PCP mutants (Vang and pk) on cytoneme morphology. No cytoneme defects are seen using fmi, fz and dsh RNAi, and the authors do not look at mutants. This leads me to wonder if the effects they see do actually involve the core PCP system, or if they involve another non-polarity role for Vang and Pk. It would be essential to determine that fmi, fz or dsh (using the PCP-specific dsh[1] allele) mutant clones do in fact have the same phenotype as Vang and pk mutants.*

It is not genetically feasible with the constructs we have to involve tracheal labeling (btl>CD8:GFP or btl>mCherry:CAAX) in all the mutants. Thus, we chose to conduct these experiments in the over-expression condition, since previous studies show that loss- and gain-of-function conditions for PCP genes have similar effects on planar polarity. Over-expression of *fz* and *fmi* in either the ASP or disc was without apparent consequence. We included these results in Figure 2—figure supplement 1. Over-expression of *dsh* in either the ASP or disc caused severe abnormality in the ASP. We agree with the reviewers that the role of *pk* and *Vang* in cytoneme-mediated signaling should not be linked to the role of these genes in planar polarity.

Reviewer #3:

*[…] Their observations are novel and very interesting, and for the most part well supported, but I do have concerns with some of the data.*

*1) Some of the reported effects on expression levels are rather subtle, yet there is no information provided as to how changes in expression levels were assessed – no quantitation or scoring criteria, no indication of numbers scored.e.g. Figure 5, Figure 6, Figure 8, Figure 9, Figure 4—figure supplement 1A-H*.

We added quantification for these images in data statistics.

*2) The claim that Tkv and Btl-containing cytonemes form in different layers is intriguing, but supported only by 3 figure panels (Figure 7), with no indication of numbers scored or quantitation. Also, the authors infer that this reflects formation or migration in Dlp vs Dally containing ecm layers, but couldn't this be tested experimentally, e.g. by double staining for Dally or Dlp and Tkv or Btl?*

We examined more than ten animals for each. Because cytonemes are sensitive to fixation, we image cytonemes in live preparation without fixation or antibody staining.

*3) The authors claim (Figure 7) the ecm near the ASP is 20 μm thick, which I find surprising. Despite drawing a dashed line for the basal surface of the disc epithelium, they don't report using any markers for the basal surface of these cells (e.g. they could use F-actin, or integrins), and I think this is needed to clarify the localizations of Dally and Dlp (e.g., is all the Dlp detected really all extracellular?).*

We estimated the distance between the collagen layer and the disc surface by the diameter of ASP (~20-30 um) which resides between them. We rewrote the text to describe it better. It is technically challenging to involve a third antibody to specifically label the basal surface of disc cells. F-actin and integrins likely also stain trachea and its associated cytonemes.

*4) The title "cytoneme navigation requires the planar polarity system", and the interpretation in the text that goes along with it, is over-stated. The authors clearly show a requirement for Pk. But Pk has also been reported to have non-PCP functions. Most notably, it’s been implicated in microtubule orientation, and vesicle transport along microtubules unrelated to PCP (e.g. Ehaideb et al. 2014), which seems more likely to account for the reduced basal secretion of glypicans than the core PCP system, whose components localize to the apical side of epithelial cells. The authors also report an effect of Vang, but if I understand Table 1 correctly, fz, dsh, and flamingo had no effect, which would be inconsistent with a role for the core PCP system. Perhaps the influence of Vang could be accounted for by an influence on the levels or localization of Pk? In any event, I think the authors should give rather more weight to the likely-hood that the requirements for Pk and Vang do not reflect a role for PCP here.*

We agree with the reviewer that the role of the PCP genes, *pk* and *Vang* in cytoneme-mediated signaling should not be linked to the role of these genes in planar polarity. We also agree that absence of cytoneme extensions in the various mutant conditions we analyzed does not directly implicate a process for path choice. These are excellent points and we have revised the manuscript accordingly. We attributed our observations to the function of Pk and Vang. As we discussed, given that PCP proteins have been implicated in ECM organization, integrin signaling, axon guidance, polarized extensions of epithelial cells, neural tube closure, directed cell movement and non-canonical Wnt signaling, it is not known if these different roles of PCP proteins are related or if they involve unrelated processes that share common components.